# FACILITATING MULTI-TURN FUNCTION CALLING FOR LLMS VIA COMPOSITIONAL INSTRUCTION TUNING

**Mingyang Chen**[1*]**, Haoze Sun**[1*]**, Tianpeng Li**[1*]**, Fan Yang**[1]**, Hao Liang**[2]**, Keer Lu**[2]**,
Bin Cui**[2]**, Wentao Zhang**[2†]**, Zenan Zhou**[1†]**, Weipeng Chen**[1]
[1]Baichuan Inc., [2] Peking University
{chenmingyang, sunhaoze, litianpeng, zhouzenan}@baichuan-inc.com
wentao.zhang@pku.edu.cn

## ABSTRACT

Large Language Models (LLMs) have exhibited significant potential in performing diverse tasks, including the ability to call functions or use external tools to enhance their performance. While current research on function calling by LLMs primarily focuses on single-turn interactions, this paper addresses the overlooked necessity for LLMs to engage in multi-turn function calling—critical for handling compositional, real-world queries that require planning with functions but not only use functions. To facilitate this, we introduce an approach, `BUTTON`, which generates synthetic compositional instruction tuning data via bottom-up instruction construction and top-down trajectory generation. In the bottom-up phase, we generate simple atomic tasks based on real-world scenarios and build compositional tasks using heuristic strategies based on atomic tasks. Corresponding function definitions are then synthesized for these compositional tasks. The top-down phase features a multi-agent environment where interactions among simulated humans, assistants, and tools are utilized to gather multi-turn function calling trajectories. This approach ensures task compositionality and allows for effective function and trajectory generation by examining atomic tasks within compositional tasks. We produce a dataset `BUTTONInstruct` comprising 8k data points and demonstrate its effectiveness through extensive experiments across various LLMs [1].

## 1 INTRODUCTION

Large Language Models (LLMs) have demonstrated remarkable capabilities across various tasks (OpenAI, 2023; Dubey et al., 2024). Beyond generating human-like text, recent studies have shown that LLMs can also call functions (i.e., use external tools) to perform specific actions or operations (Qin et al., 2023; Patil et al., 2023; Mu et al., 2024; Liu et al., 2024b). This ability enhances LLM performance, such as retrieving information from external knowledge bases to reduce hallucinations (Schick et al., 2023; Gao et al., 2023; Zhao et al., 2024; Wang et al., 2024c). Moreover, LLM-based agents can interact with numerous external APIs through function calls, providing standardized interfaces that increase their utility and versatility in real-world applications (Chen & Li, 2024; Wang et al., 2024a; Zeng et al., 2024).

Existing research on aligning LLMs for function calling predominantly focuses on a single-turn approach, primarily constructing instruction data to teach and evaluate them on selecting appropriate functions and providing the correct arguments (Patil et al., 2023; Liu et al., 2024b). While it is important for LLMs to learn how to understand and **_use functions_**, these studies often overlook the crucial ability to **_plan with functions_**. Many real-world user queries are complex and cannot be resolved in a single step. For example, "List the flight schedule from London to Edinburgh" may be a single-step task since simply retrieving exact information can complete it, while "Book me the first flight from London to Edinburgh" requires calling multiple functions sequentially: first retrieving the flight schedule and finding the first one, then booking a ticket for that flight. Real-world complex

---

[*]Equal Contribution
[†]Corresponding Author
[1]The data is available at `https://github.com/PKU-Baichuan-MLSystemLab/BUTTON`

tasks are inherently compositional (Press et al., 2023; Hayati et al., 2024; Hu et al., 2024), requiring LLMs not only to invoke functions but also to decompose these tasks into manageable steps and plan the sequence of function calls. Therefore, in this study, we focus on constructing an instruction-tuning (Ouyang et al., 2022; Longpre et al., 2023; Lin et al., 2024) dataset where the inputs are complex compositional queries and the outputs are their decompositions into multi-turn function calls, addressing real-world scenarios of function calling for LLMs via *Compositional Instruction Tuning*.

However, acquiring such data from existing sources is unrealistic. First, selecting and identifying instructions in a compositional manner is challenging (Hayati et al., 2024; Hu et al., 2024), and finding instructions paired with their corresponding functions is even more difficult (Shen et al., 2023; Schick et al., 2023; Patil et al., 2023). Additionally, we need a "solution" in the form of labeled multi-turn function calls that align with the compositional instructions based on the given functions. All these factors make it difficult to obtain such data without extensive manual annotation. Recently, synthetic data has emerged as a promising solution to the lack of manually curated data, with data being created through advanced generative LLMs using tailored processes and simulations (Liu et al., 2024a). Compared to synthetic data in general domains (Xu et al., 2023; Lou et al., 2023; Xu et al., 2024a), data in our scenarios must consider the following challenges: 1) How to ensure the compositionality of generated instructions so that they are complex, reasonable and solvable; 2) How to ensure the compatibility of an instruction with its functions; 3) How to simulate high-quality multi-turn function calling trajectories without human supervision.

To address these challenges, we propose BUTTON, a "**B**ottom-**U**p **T**hen **T**op-d**O**w**N**" pipeline for generating synthetic compositional instruction tuning data to enhance the multi-turn function calling abilities of LLMs. In the "*Bottom-Up*" phase, we begin by generating atomic tasks from general real-world scenarios. These tasks are designed to be simple, clear, and executable in a single step without the need for planning. Compositional tasks are constructed based on atomic tasks using two heuristic strategies: Sequential Composition and Parallel-then-Sequential Composition. Although straightforward, these two composition strategies, combined with the entire pipeline, can generate diverse compositional instructions. Then, corresponding functions are generated based on compositional tasks with the conscious of their atomic tasks. During "*Top-Down*" phase, we set up a multi-agent environment where the human, assistant and tools are simulated by generative LLMs steered by specifically curated system prompts, where tool agents are simulated according to previous generated function definitions. The trajectory of multi-turn function calling, initiated by a user and involving interactions between an assistant and tools, is collected based on this simulated environment. Finally, the collected trajectories, along with their corresponding function definitions, are filled into a predefined prompt template to serve as instruction tuning data for LLMs. The bottom-up procedure, rather than generating complex tasks directly, ensures compositionality of instructions (challenge 1). Generating functions with an awareness of the atomic tasks within compositional tasks makes these functions more general and suitable for fine-grained sub-tasks, rather than being monolithic (challenge 2). Using multi-agents to simulate the trajectories enhances their quality, and examining the sub-tasks for compositional tasks also guides the agents toward effective decomposition and planning with functions (challenge 3). Based on BUTTON, we collected a compositional instruction tuning dataset called BUTTONInstruct, consisting of 8k high-quality data points labeled with multi-turn function call trajectories. We demonstrate that LLMs fine-tuned with BUTTONInstruct show improved performance on multi-turn function calling benchmarks.

## 2 METHOD

In this section, we introduce the details of our method for constructing compositional instruction-tuning data for multi-turn function calling tasks. This framework consists of two stages: 1) bottom-up instruction construction, and 2) top-down trajectory generation. We refer to this framework as the "Bottom-Up Then Top-dOwN" pipeline, denoted as BUTTON.

During the bottom-up instruction construction phase, we begin by gathering a variety of real-world scenarios. Drawing from these scenarios, we proceed to construct a series of atomic tasks, each as simple as possible and capable of being completed in one step. Compositional tasks will be evolved from such atomic tasks. Lastly, for each compositional task, we further generate function definitions that are likely to be called in this task. The term "bottom-up" signifies our approach of constructing

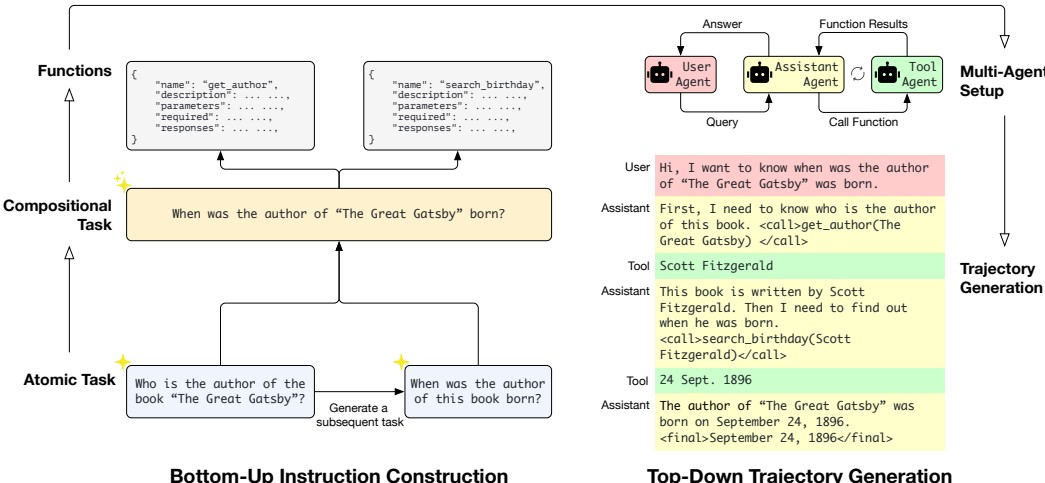

Figure 1: Overview of our bottom-up then top-down pipeline.

comprehensive compositional tasks with their corresponding functions, starting from the foundation of simple atomic tasks, which can be shown in the left part of Figure 1.

During the top-down trajectory generation phase, we collect multi-turn function calling interaction trajectories for compositional tasks and functions that were constructed earlier. This step simulates the usage process of functions in each task, providing supervision data that guides the LLMs in learning how to call functions in multi-turn manners. The term "top-down" indicates that from a compositional task with corresponding functions, we simulate and gather the interaction trajectories that demonstrate the breakdown of complex tasks and the invocation of corresponding functions in a multi-turn manner, as depicted in the right part of Figure 1. In essence, the bottom-up instruction construction is a process of composition, while the top-down trajectory generation is a process of decomposition.

**Definitions** We begin by collecting real-world scenarios $\mathcal{C} = \{c_i\}$ and transforming them into atomic tasks $\mathcal{A} = \{a_i\}$. For compositional tasks requiring multiple function calls, we generate related atomic tasks from a defined atomic task $a_i$, forming a sub-task set $\mathcal{S}_i$. This set is used to compose a compositional task $c_i$, represented as $\mathcal{C} = \{(c_i, \mathcal{S}_i)\}$. Functions are then constructed based on these tasks, forming instruction tuples $\mathcal{I} = \{(c_i, \mathcal{F}_i, \mathcal{S}_i)\}$, where $\mathcal{F}_i$ are a set of functions for task $c_i$. These functions are defined by their descriptions, allowing us to simulate interactions without actual implementations. The task breakdown $\mathcal{S}_i$ is retained to enhance multi-agent inter-action for trajectory generation, without being disclosed in the final data. After obtaining $\mathcal{I}$, we generate user, assistant, and tool interactions, resulting in data $\mathcal{D} = \{(\mathcal{F}_i, t_i)\}$, where $t_i$ is the collected trajectory. These function definitions and trajectories are integrated into a prompt template for instruction-tuning data.

## 2.1 BOTTOM-UP INSTRUCTION CONSTRUCTION

In this section, we detail the procedures for constructing instructions using a bottom-up approach. The process begins with simple scenarios, from which we generate tasks at the atomic level. These atom tasks are then evolved to create more complex, composite tasks along with their corresponding functions.

**Scenario Collection** To ensure our generated tasks are grounded in everyday experiences and not meaningless, we first extract a series of real-world scenarios from existing datasets that focus on function calling for LLMs (glaiveai, 2023; Qin et al., 2023). Such scenarios can be a concise overview, like "book a flight" or "ordering meals". We then conduct a deduplication operation on the collected scenarios. This involves using sentence embeddings to calculate scenario similarities, and setting a specific threshold to filter out similar ones (Xiao et al., 2023). Furthermore, to expand our scenario collection, we also attempt to generate new scenarios from existing ones by altering

Table 1: Examples of compositional task construction.

| **Sequential Composition** | |
|---|---|
| Initial Atomic Task: | Who is author of the book "The Great Gatsby"? |
| Subsequent Task: | When was the author of this book born? |
| Composition Task: | When was the author of the book "The Great Gatsby" born? |
| **Parallel-then-Sequential Composition** | |
| Initial Atomic Task: | Give me the flight schedule from London to Edinburgh today. |
| Parallel Task: | Find the every hour weather forecast in Edinburgh today. |
| Subsequent Task: | What is the weather condition when the first flight arrives? |
| Composition Task: | I am in London, and I want to know the weather condition when the first flight arrives in Edinburgh today. |

their actions or subjects. Details of seed data for scenario extraction, prompts for extracting and expanding scenarios can be found in Appendix A.1.

**Atomic Task Construction**    Based on the collected scenarios, we construct atomic tasks, each of which can be considered as a straightforward problem, query or instruction. We anticipate that these atomic tasks should be simple, clear, and don't need complex planning for human to solve. Such atomic tasks are able to serve as atomic units for constructing complex compositional tasks. In designing the prompts for transforming collected scenarios into atomic tasks, we focus on the following three aspects:

- **Reasonable**: The atomic task should be realistic, reasonable, and representative of tasks frequently encountered in the real world.

- **Self-contained**: The atomic task should be solvable based on the information it provides. It must contain sufficient details and information necessary for calling functions.

- **Function-agnostic**: During the articulation of the atomic task, we do not consider the specific functions that can be employed to solve this task. The task should not mention any specific function or solution.

Note that despite having numerous descriptions of our anticipated atomic tasks, we do not provide a strict definition of atomic tasks, nor of the following compositional tasks. We use specific prompts and the powerful instruction-following capabilities of cutting-edge LLMs to ensure that the generated data aligns with our general expectations. The carefully crafted prompt can be found in Appendix A.2.

**Compositional Task Construction**    To enhance the capability of multi-turn function calling for LLMs, it is essential to construct compositional tasks that require multiple interactions to be resolved. Starting with the atomic tasks created in the previous step, we develop compositional tasks using two heuristic strategies: "Sequential Composition" and "Parallel-then-Sequential Composition". The principle behind sequential composition is to start with an atomic task and generate a subsequent task that needs to be solved based on the result of the first, combining them into a new compositional task. For parallel-then-sequential composition, we begin by generating a task that can be solved in parallel with the atomic task. Then, a subsequent task is generated based on the results of the first two tasks, and they are all composed together. Examples can be found in Table 1. Prompts for implementing these two strategies can be found in Appendix A.3. Although these two heuristic strategies seem simple, they do not compromise the diversity of the dataset collected using our method. For instance, parallel function calls are not limited to the first turn; multiple functions may be generated for a single atomic task in subsequent function generations. Details on the data diversity can be found in Sec. 2.3. To ensure the quality of the generated compositional tasks, we filter out any that are inconsistent with their original atomic tasks. The filtering criterion is as follows: since the quality of atomic tasks is more controllable, we filter compositional tasks by checking whether each one can be completed by its atomic sub-tasks. This allows us to filter out low-quality compositional tasks. The task filtering prompt can be found in Appendix A.3.

**Function Generation** After constructing the compositional tasks, we generate functions that are likely to be called in these tasks. This differs from most previous works, which first collect functions and then generate tasks based on these collected functions (Patil et al., 2023; Liu et al., 2024b). Our task-generation procedure is function-agnostic, which we believe allows for the construction of more realistic tasks, rather than those based solely on specific functions. Using the aforementioned methods, we have constructed a series of compositional tasks for which we know the corresponding sub-tasks (i.e., the breakdown of the complex compositional task). These task breakdowns can be used as hints for function generation. This represents the advantage of our bottom-up instruction construction method, where we can examine the decomposition of the compositional task and generate corresponding functions. During the generation of function definitions, we mainly focus on the following points:

- **Descriptive**: The name and description of the function should be illustrative to aid in distinguishing different functions. The input arguments and output returns should also be clear since we not only use these definitions to but also use them to simulate corresponding functions.

- **General**: The function should possess a level of generality that enables its use for future tasks as well. In the real world, a function is more likely to be constructed for a frequently encountered atomic task rather than a highly specific one. For instance, a function `get_weather(city)` is more likely to be utilized than `get_weather_in_london()`.

- **Consistency**: As we need to generate multi-turn function calling interaction trajectories in later steps, the input arguments and output results of these functions should maintain consistency. For example, if two functions will be called sequentially, the output of the first should either align with or constitute a part of the input for the second function, regardless of the varying parameter terminologies.

The generated function definitions include five main fields: name, description, parameters, responses, and required. The name indicates the function name, while description details its usage and capabilities. Parameters and responses cover the input and output, including the type and description of each argument. The required field lists necessary input parameters. We allow flexibility in mapping sub-tasks to functions; a sub-task may require zero, one, or multiple functions. If a sub-task involves logic, comparison, set operations, or calculations manageable by language models, no function is needed. For more details on the format and prompts, see Appendix A.4.

## 2.2 TOP-DOWN TRAJECTORY GENERATION

After obtaining the compositional tasks and their corresponding functions through the bottom-up instruction construction method, we create multi-turn function calling interaction trajectories. These trajectories simulate how LLMs use these functions and serve as supervision data, teaching LLMs to perform multi-turn function calls.

**Multi-agent Setup** How to effectively simulate the multi-turn function calling interaction process is the key to collecting high-quality interaction trajectories. In our framework, we set up a multi-agent environment in which each agent simulates a specific role during the multi-turn function calling interactions, as shown in the right part of Figure 1. We design three types of agents to simulate the interaction process: the user, the assistant, and the tool. The user agent initiates the interaction and provides the query to the assistant agent based on a specific compositional task. The assistant agent decomposes the task into sub-tasks and calls the corresponding functions to address these sub-tasks. The tool agent simulates the specific implementations of a function. It's important to note that a tool agent simulates a specific function based on its definition. We do not implement the actual functionality of the function as we merely require reasonable feedback from the tool agents to advance the interaction. Specifically, the assistant agent is aware of the available tools, compositional tasks, and the task breakdown. Since our compositional tasks are constructed in a bottom-up manner, the breakdown of these tasks naturally comprises their atomic components. For each function, we establish a tool agent based on its definitions. The behavior of the agents is steered by their system prompts, and the details of system prompts are listed in Appendix A.5.

**Interaction Trajectory Collection.** After setting up the multi-agent environments, given a specific compositional task and corresponding functions, we tailor prompts for each agent. The interaction trajectory begins with the user agent. The assistant agent reviews the task, decides which

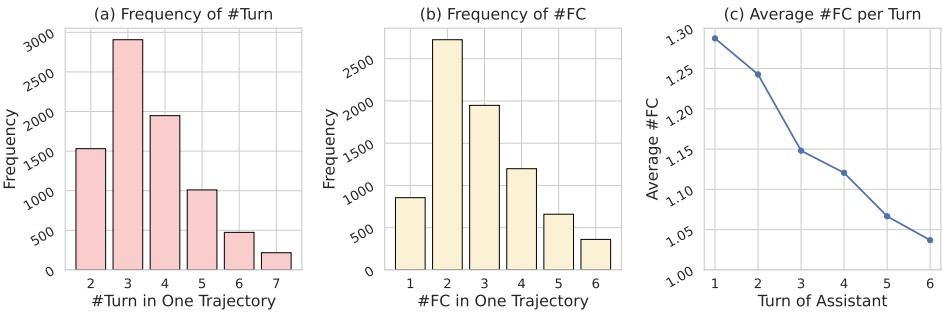

Figure 2: Statistic investigation on our collected data `BUTTONInstruct`.

functions to call, and determines the function parameters. The assistant's response includes: first, observations and thoughts in free text, similar to the ReAct (Yao et al., 2023) format; second, a specified function call. This function call is parsed as an action to invoke the corresponding tool agent and obtain simulated function call results. Finally, when the assistant has the final answer to the user's question, it invokes the function that provides the final response to the user.

## 2.3 DATASET COLLECTION VIA BUTTON

**BUTTONInstruct** Based on the aforementioned pipeline, we leverage the cutting-edge LLM, GPT-4o[2], to generate data at each step of bottom-up instruction construction, progressing from simple scenarios to compositional tasks and functions, and conducting trajectory generation as agents. We ultimately collect 8,000 multi-turn function calling data points, `BUTTONInstruct`, each containing several entries, including content with roles such as 'system', 'user', 'assistant', and 'tool'. The available functions for each user question are listed in corresponding system prompt. More examples of finally generated data can be found in Appendix B.1.

**Parallel Function Calling** Furthermore, in our collected data, we consider scenarios involving parallel function calling. If multiple functions can be called independently, they can be executed in parallel. The effectiveness of this parallel calling is discussed in Section 3.3. To control the behavior of conducting parallel calling, we use different system prompts to guide whether parallel calling should be executed. Details on how to construct data for aligning LLMs with the corresponding calling behaviors can be found in Appendix B.2.

**Data Diversity** To demonstrate the diversity of the collected `BUTTONInstruct`, we analyzed the distribution of the total number of assistant turns (#Turn), as well as the number of function calls (#FC) throughout the entire trajectory and at each step. Figure 2 presents key statistics of our dataset. In Figure 2(a), the frequency distribution of the total number of turns shows that most data points involve three or more turns of assistant responses. We plotted the distribution of the total number of function calls for each trajectory in Figure 2(b) and found that most data points contain more than two function calls. By plotting the average number of function calls per turn in Figure 2(c), we observed that every turn averages more than one function call. These findings indicate the diversity in our collected data and demonstrate that, even with simple heuristic strategies to generate compositional tasks from atomic tasks, the final collected data are diverse across different aspects. Furthermore, more details about the distribution of the functions in the `BUTTONInstruct` are provided in Appendix B.3.

## 3 EXPERIMENTS

To evaluate the effectiveness of our multi-turn function calling data `BUTTONInstruct` collected via our proposed `BUTTON` pipeline, we train two series of open-source LLMs of different sizes: Llama3-8B, Llama3-70B (Dubey et al., 2024), Qwen2-7B, and Qwen2-72B (Yang et al., 2024a). We denote models that have been instruction-tuned using `BUTTONInstruct` with the suffix '-BUTTON'. In our experiments, we primarily focus on the following questions: 1) *Q1*: Can our

---

[2]gpt-4o-2024-05-13 from `https://platform.openai.com/docs/models/gpt-4o`

proposed BUTTON approach enhance multi-turn function calling abilities compared to existing instruction-tuned models designed for general purposes? 2) *Q2*: Are the bottom-up and top-down procedures effective and necessary? 3) *Q3*: What is the influence of data size and parallel calling on model performance?

## 3.1 EXPERIMENTS SETUP

**Benchmarks.** We evaluate performance using two benchmarks, GTA and Tool-Query. **GTA** (Wang et al., 2024b), a benchmark for General Tool Agents, consists of 229 human-crafted queries designed to reflect real-world tasks. The queries span a total of 14 real-world deployed tools (i.e., functions) across the perception, operation, logic, and creation categories. Each query is accompanied by one or two authentic image files and the LLM is tasked with solving the queries based on the multimodal context and user queries. The tasks in this benchmark require multiple steps to solve and necessitate LLMs to reason about suitable tools and plan the solution steps. More details and examples about this benchmark can be found in Appendix B of Wang et al. (2024b). **Tool-Query** (Ma et al., 2024) is a tool-using environment in the domains of weather, movies, and academia. It consists of 60 tasks requiring complex multi-round interactions with corresponding tools. There are 18, 14, and 7 tools (i.e., functions) in the weather, movie, and academia environments respectively, which are developed by corresponding real-world APIs and databases. This benchmark not only contains annotated final answers but also intermediate subgoals, which makes it easier to evaluate the process of function calling during multi-turn interactions. Tasks are also labeled as hard or easy based on the number of subgoals, with a count of 4 in this benchmark. More details and examples of Tool-Query can be found in Appendix F.4.1 of Ma et al. (2024).

**Evaluation Metrics.** For each benchmark, we follow their original evaluation strategies and metrics. For **GTA**, there are two modes and nine metrics. In the Step-by-Step Mode, the model is provided with steps 1 to $n$ from a set of human-labeled function calling chains, and it is tasked with predicting the function call in the $n + 1$ step. This demonstrates performance in a fine-grained way. In the End-to-End Mode, the model initiates its function calling process based solely on the user's question and proceeds until it arrives at the answer. This reflects the performance of the model in real-world applications. Four metrics are used during the step-by-step mode evaluation: Instruction Accuracy (*Inst.*) is the accuracy of executing without errors, which indicates that the model knows how to follow the instruction to conduct a function call; Tool-selection Accuracy (*Tool.*) and Argument Accuracy (*Arg.*) denote the accuracy of selecting tools and predicting arguments respectively; Summary Accuracy (Summ.) denotes the model's ability to summarize and derive the answer based on all previous steps. During end-to-end mode, we show the F1 scores of tool selection on perception (*P.*), operation (*O.*), logic (*L.*), and creativity (*C.*) tasks, and the final answer accuracy (*Ans.*). The final answer accuracy is only calculated solely on queries with pure text answers, excluding image generation queries. For **Tool-Query**, Grounding Accuracy (*G.A.*) is used to denote the accuracy of generating valid, executable function calls. Process Rate is used to evaluate the completion proportion of subgoals during the handling of complex user queries and the subgoals are labeled in this dataset. Furthermore, Success Rate is the accuracy of the final answer. More details of metric calculation can be found in Appendix C.1.

**Baselines.** We showcase the performance on cutting-edge API-based LLMs, including GPT-4o, GPT-4-Turbo, and GPT-3.5-Turbo. Furthermore, since our collected data BUTTONInstruct is adapted for tuning the Llama3 and Qwen2 base models, we also use their original instruction-tuned versions as baselines, including Llama3-8B-Instruct, Llama3-70B-Instruct, Qwen2-7B-Instruct, and Qwen2-72B-Instruct (Dubey et al., 2024; Yang et al., 2024a).

**Implementation.** We conduct full-parameter supervised fine-tuning for all base models, with a maximum sequence length of 8,192. The models are trained on 4×8 NVIDIA H800 GPUs. In addition, we randomly select 100,000 general instruction tuning data from OpenHermes-2.5 (Teknium, 2023) and mix them with BUTTONInstruct to align the models' basic instruction-following capabilities. For implementation details, please refer to Appendix C.2.

## 3.2 MAIN RESULTS

As shown in Table 2 and Table 3, LLMs tuned with compositional instruction tuning data constructed by our proposed BUTTON method perform better than their corresponding original instructed versions. For smaller models, the improvements are significant. For example, we improved the an-

swer accuracy of Llama3-8B from 1.4% to 30.5%. The low performance of Llama3-8B-Instruct on this metric is consistent with the findings in the original benchmark paper. Notably, the results of Llama3-80B-BUTTON and Qwen2-72B-BUTTON are comparable to GPT-4o, demonstrating the effectiveness of the data collected through our proposed data collection pipeline.

Table 2: Main results(%) on GTA. Accuracy of *Inst.*, *Tool.*, *Arg.*, and *Summ.*. F1 score of *P.*, *O.*, *L.*, and *C.*. *Ans.* is the final answer accuracy. **Bold** numbers highlight better performance between the original instruction model and our tuned versions, while underlined numbers denote the best performance across all models.

| Model | Step-by-Step Mode | | | | End-to-End Mode | | | | |
|---|---|---|---|---|---|---|---|---|---|
| | *Inst.* | *Tool.* | *Arg.* | *Summ.* | *P.* | *O.* | *L.* | *C.* | *Ans.* |
| GPT-4o | 90.0 | 70.3 | 38.6 | 72.9 | 76.4 | 88.2 | 84.8 | 90.0 | 46.0 |
| GPT-4-Turbo | 84.6 | 60.6 | 34.3 | 73.5 | 51.8 | 87.7 | 61.8 | 86.1 | 30.6 |
| GPT-3.5-Turbo | 64.8 | 33.1 | 22.4 | 66.2 | 60.2 | 48.4 | 69.0 | 95.8 | 18.3 |
| Llama3-8B-Instruct | 70.9 | 23.6 | 2.2 | 42.9 | 23.5 | 18.5 | 23.6 | 24.6 | 1.4 |
| Llama3-8B-BUTTON | **90.7** | **63.4** | **32.3** | **65.6** | **84.2** | **76.5** | **73.5** | **88.9** | **30.5** |
| Qwen2-7B-Instruct | 59.1 | 28.5 | 3.9 | 54.8 | 39.1 | 65.4 | 56.1 | 72.7 | 13.1 |
| Qwen2-7B-BUTTON | **89.4** | **62.5** | **30.7** | **63.0** | **80.3** | **83.5** | **82.6** | **89.2** | **27.3** |
| Llama3-70B-Instruct | 75.2 | 46.7 | 22.2 | 68.4 | 67.9 | 83.8 | 71.0 | 95.9 | 40.1 |
| Llama3-70B-BUTTON | **96.4** | **73.6** | **38.1** | **70.5** | **84.9** | **96.2** | **89.5** | **96.1** | **43.5** |
| Qwen2-72B-Instruct | 73.4 | 49.2 | 17.9 | **73.9** | 46.5 | 67.5 | 46.7 | 64.3 | 27.3 |
| Qwen2-72B-BUTTON | **91.9** | **69.3** | **38.1** | 71.5 | **85.0** | **87.4** | **86.7** | **91.4** | **45.7** |

For Tool-Query, LLMs tuned with our data `BUTTONInstruct` demonstrate improved performance across various metrics. We observe that even though the grounding accuracy for both the original instruction models and those tuned with our dataset is quite high (near or above 95%), our models achieve better process and success rates. This suggests that the challenge lies not only in using functions correctly but also in effectively planning with them. Overall, from the main results from GTA and Tool-Query, we can conclude that `BUTTONInstruct` can align LLMs with better ability on multi-turn function calling, and `BUTTON` is effective on such tasks.

Table 3: Main results(%) on Tool-Query. *G.A.* indicate grounding accuracy. Process Rate and Success Rate are presented for *Easy*, *Hard* and *ALL* test samples. **Bold** numbers highlight better performance between the original instruction model and our tuned versions, while underlined numbers denote the best performance across all models.

| Model | G.A. | Process Rate | | | Success Rate | | |
|---|---|---|---|---|---|---|---|
| | | *Easy* | *Hard* | *All* | *Easy* | *Hard* | *All* |
| GPT-4o | 92.3 | 83.2 | 70.6 | 76.5 | 50.0 | 31.3 | 40.0 |
| GPT-4-Turbo | 95.4 | 80.7 | 78.9 | 79.8 | 50.0 | 34.4 | 41.7 |
| GPT-3.5-Turbo | 93.6 | 54.9 | 43.2 | 48.7 | 3.6 | 9.4 | 6.7 |
| Llama3-8B-Instruct | 96.7 | 55.1 | 42.0 | 48.1 | 10.7 | 0.0 | 5.0 |
| Llama3-8B-BUTTON | **97.4** | **72.9** | **54.8** | **63.2** | **50.0** | **21.9** | **35.0** |
| Qwen2-7B-Instruct | **97.0** | 66.7 | 46.8 | 56.0 | 32.1 | **15.6** | 23.3 |
| Qwen2-7B-BUTTON | 95.5 | **69.5** | **59.0** | **63.9** | **42.9** | **15.6** | **28.3** |
| Llama3-70B-Instruct | **95.6** | 80.8 | 61.5 | 70.5 | 42.9 | 21.9 | 31.7 |
| Llama3-70B-BUTTON | 94.0 | **85.2** | **77.2** | **80.9** | **71.4** | **46.9** | **58.3** |
| Qwen2-72B-Instruct | 95.8 | 83.7 | 72.2 | 77.6 | 50.0 | 34.4 | 41.7 |
| Qwen2-72B-BUTTON | **98.4** | **85.5** | **77.0** | **81.0** | **71.4** | **46.9** | **58.3** |

## 3.3 FURTHER ANALYSIS

**Ablation Study** To evaluate the effectiveness of the "bottom-up instruction construction" and "top-down trajectory generation" procedures in BUTTON respectively, we conduct an ablation study by simplifying these two procedures into single direct generation steps using one prompt with generative LLMs, instead of our curated prompts and procedures. Specifically, to simplify the bottom-up procedure, we instruct generative LLMs to directly generate a compositional complex task based on a given scenario. As a result, no sub-tasks are revealed for the subsequent function generation and trajectory generation steps. To reduce the top-down procedure, we do not set up the multi-agent environment. Instead, we use a monolithic prompt that instructs the LLMs to act as the user, assistant, and tools, generating trajectories based on previously constructed tasks with functions. Refer to Appendix C.3 for details of prompts. The results of the ablation study on Tool-Query are shown in Figure 3. We perform a comparison using the aforementioned two ablation settings (i.e., w/o Buttom-Up and w/o Top-Down), and the default setting (i.e., w/ BUTTON) on

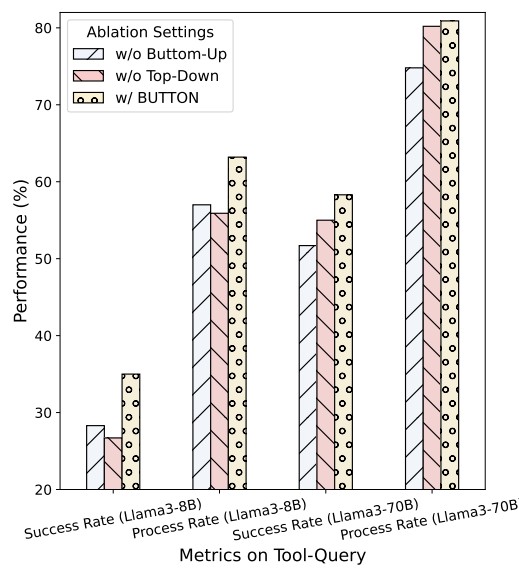

Figure 3: Performance comparison in ablation settings for metrics Tool-Query.

Llama3-8B and Llama3-70B. By calculating the relative performance degradation, we found that the smaller model was influenced more, with average relative degradations on process rate and success rate of 16.1%, and that decline for Llama3-70B is 6.4%. The average relative success rate decline across both model sizes and ablation settings is 15.0%, while the process rate decline is 7.5%. This suggests that models tuned without BUTTON can partially execute correct functions contributing to the process rate, but their limited planning ability prevents them from effectively achieving the final answer, leading to a greater decline in the success rate.

Table 4: Results(%) on Tool-Query with Llama3-8B tuned with varying data sizes.

| #Data | Process Rate | | | Success Rate | | |
|---|---|---|---|---|---|---|
| | *Easy* | *Hard* | *All* | *Easy* | *Hard* | *All* |
| 2,000 | 67.5 | 46.9 | 56.5 | 35.7 | 9.4 | 21.7 |
| 4,000 | 73.1 | 45.8 | 58.5 | **50.0** | 12.5 | 30.0 |
| 6,000 | **74.6** | 51.5 | 62.3 | 46.4 | 15.6 | 30.0 |
| 8,000 | 72.9 | **54.8** | **63.2** | **50.0** | **21.9** | **35.0** |
| AGR | 13.5 | 32.8 | 18.4 | 2.7 | 5.5 | 3.8 |

**Data Scaling** In this section, we investigate the influence of data size on compositional instruction tuning data constructed by BUTTON. We vary the data size from 2,000 to 8,000 and tune Llama3-8B with the corresponding data. As shown in Table 4, we present the results on Tool-Query with varying data sizes. By calculating the average growth rate (AGR), we found that as the data size increases, the process rate and success rate increase accordingly, with AGRs of 13.5% and 2.7%, respectively. Furthermore, we found that the improvement on hard samples is greater than on easy samples, indicating that our constructed data can effectively enhance performance on multi-turn function calling as the data size increases, particularly for tasks that require more turns of function calling.

**Efficiency on Parallel Calling** As described in Section 2.3, we have mentioned that our tuned model can conduct parallel function calling when multiple functions can be called independently within a single turn. This capability can significantly enhance performance when operating under restricted step conditions. By turning off the parallel calling ability by changing the system prompt, we found that performance on different metrics is affected. For example, the success rate of Llama3-8B-BUTTON on Tool-Query decreases from 35.0% to 28.3%, and the process rate decreases from 63.2% to 58.7%. This indicates the effectiveness of parallel calling and our proposed BUTTON method. More cases about comparisons of parallel calling can be found in Appendix C.4.

## 4 RELATED WORK

**Synthetic Data** Data has been a critical driving force behind the success of large language models (LLMs). Recent advancements in these models owe much to the availability of large-scale, diverse, and high-quality datasets (Dubey et al., 2024). However, obtaining such datasets is both resource-intensive and expensive, posing significant challenges to researchers and practitioners (Xu et al., 2023; 2024a; Luo et al., 2023; Yang et al., 2024b). To address these limitations, generating synthetic data has emerged as a promising alternative for creating scalable and high-quality datasets. Researchers have explored various techniques to improve the quality, diversity, and utility of synthetic data across different domains (Liu et al., 2024a; Wang et al., 2024d; Wei et al., 2023; Lou et al., 2023; Yang et al., 2023; 2024c). Recent studies have also focused on synthetic data for complex and compositional instructions. Chain-of-Instruct (Hayati et al., 2024) proposed a pipeline for building instructions composed of subtasks, allowing LLMs to solve compositional tasks step-by-step in an explainable manner. Sequential Instruction Tuning (Hu et al., 2024) introduced a data construction pipeline for automatic instruction augmentation, where intermediate tasks are seeded from a single-task instruction. However, existing works do not focus on constructing compositional instructions for multi-turn function calling tasks, where identifying, invoking, and planning with functions are all necessary.

**Function Calling** The ability to call functions or use tools effectively has become a significant aspect of instruction-tuning large language models (LLMs) (Wang et al., 2024a; Zeng et al., 2024; Shen et al., 2023; Xu et al., 2024b). This functionality is crucial as it enhances the models' ability to tackle complex tasks by enabling modular, structured, and interactive workflows (Qiao et al., 2024; Shen et al., 2024). Such capabilities expand the potential of LLMs beyond language generation, empowering them to dynamically address practical challenges that require integrating computational or domain-specific tools. Recent efforts have focused on creating and curating datasets to enhance LLMs' function-calling competencies. Patil et al. (2023) collected APIs from repositories such as TorchHub, TensorHub, and HuggingFace, subsequently generating synthetic user prompts for each API using the Self-Instruct framework (Wang et al., 2023). Similarly, Qin et al. (2023) expanded upon this work by incorporating additional API data from RapidAPI. They generated diverse instruction datasets involving a variety of function-calling contexts, covering numerous interaction scenarios. However, these approaches face significant challenges: many API descriptions are unclear, and a considerable number of API calls fail due to availability issues or runtime errors (Guo et al., 2024), complicating the generation of accurate solution path annotations using LLMs. To address some of these challenges, Liu et al. (2024b) proposed APIGen, a multi-stage pipeline for generating diverse datasets tailored to function-calling tasks. This method employs a rigorous verification process to improve data quality but is limited in scope, focusing predominantly on single-turn function-calling scenarios.

## 5 CONCLUSION

In this work, we address the importance of multi-turn function calling in the field of LLMs by focusing on their ability to plan with functions, rather than merely use them. We introduce `BUTTON`, a novel "bottom-up then top-down" pipeline for generating synthetic compositional instruction tuning data. This approach effectively tackles the challenges of ensuring compositionality of tasks, generating compatible function, and high-quality multi-turn function calling trajectories without human supervision via the curated prompts and procedures during the pipeline. Our methodology, which includes the generation of atomic tasks, compositional task construction, function generation, and a multi-agent simulation environment, has resulted in the creation of `BUTTONInstruct`, a dataset of 8,000 high-quality data points labeled with multi-turn function call trajectories. The effectiveness of this approach is demonstrated by the improved performance of LLMs fine-tuned with `BUTTONInstruct` on multi-turn function calling benchmarks. However, although our method has been empirically proven effective through experiments and analysis, the quality of the data currently relies on our prompts and procedures, with no additional verification steps applied. In future work, we will implement more curated data verification or filtering stages to enhance the quality of the synthetic data. Furthermore, we may focus on extending the pipeline to align LLMs with more real-world applications, including embodied AI, where tool use, complex reasoning, and planning need to be integrated to complete more complex tasks.

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

# A  PROMPT DETAILS

## A.1  SCENARIO COLLECTION

In our work, the seed data for scenario extraction is derived from glaive-function-calling-v2 (glaiveai, 2023) and ToolLLama datasets (Qin et al., 2023).

Here are prompts for extracting and expanding scenarios. The placeholders {conversation} and {scenario} are used to fill in a conversation for extracting a scenario and a scenario for being modified, respectively.

---

**Prompt for Extracting Scenarios**

```
Please analyze the conversation below between a user and an
    assistant bot and identify the general life scenario it
    represents. Provide a concise overview of the scenario type,
    such as 'booking flights' or 'ordering meals'. Avoid
    mentioning specific details like numbers or items. Your
    response should be a description of the scenario without
    additional commentary, and should not exceed 10 words.

Conversation:
{conversation}

Concise Overview of the Scenario:
```

---

**Prompt for Expanding Scenarios**

```
Based on the provided daily scenario, creatively generate a new
    and entirely different scenario. The new scenario must meet
    the following requirements:
1. You may alter the action or subject of the original scenario.
2. The new scenario should differ substantially from the
   original.
3. Ensure the new scenario is realistic and feasible within a
   daily life context.
4. Retain the same format as the original scenario.
5. Limit your response to 10 words and present the new scenario
   in a single sentence.

Original Scenario:
{scen}

Modified Scenario:
```

---

## A.2  ATOMIC TASK CONSTRUCTION

Here is the prompt for generating an atomic task from a scenario. The placeholder {scenario} will be substituted with a collected scenario when generating a specific task.

---

**Prompt for Atomic Task Construction**

```
You are training a model that can take a user's task description
    or query, and available functions as input, and generate a
    sequence of function calls to accomplish the task. Currently,
    you are generating basic atom tasks. Given a general life
```

---

```
    scenario as the context, please generate a basic atom task
    that can be accomplished in one step.

Requirements of the task:
1. The task should be a reasonable real life task based on the
   given scenario, and can be accomplished in one step.
2. If you mention some information, criteria or constraints in
   the task, please give the details of these information,
   criteria or constraints. Do not assume the model has access
   to your personal information or prior knowledge, and it does
   not have chance to ask you for clarification.
3. Please give enough details and make the task description as
   specific as possible, so the model can make deterministic
   function calls with deterministic arguments. Do not include
   any ambiguous or vague information.
4. Do not mention specific tools or functions in the task
   description, and do not propose solutions, hints, or project
   outcomes.
5. Limit the task description to 30 words, and avoid using
   adjectives and ambiguous words.

Given Scenario:
{scenario}

Please give your response in one line directly, without any
    extra notation or format:
```

## A.3 COMPOSITIONAL TASK CONSTRUCTION

Following two prompts are used for constructing compositional tasks from atomic tasks, including sequential composition and parallel-then-sequential composition strategies.

**Prompt for Sequential Composition**

```
You are training a model that can take a user's task description
    or query, and available functions as input, and generate a
    sequence of function calls to accomplish the task. Currently,
    you are generating complex tasks for model training. Given a
    task, you need to add a subsequent task for this given task
    to make a more complex task.

The requirements for the subsequent task are as follows:
1. The subsequent task should use the output of the given task
   as input.
2. The subsequent can only be conducted after the given task has
   been completed.
3. The subsequent task and the given task can form a new
   compositional task, and composing them can make a more
   complex multi-step task.

## Examples:
### Given Task: Give me a list of all the pets.
### Subsequent Task: What is the most common kind of pet in the
    list?
### Composition Task: Check the most common kind of pet in the
    list of all the pets.
```

```
### Given Task: Who is author of the book "The Great Gatsby"?
### Subsequent Task: When was the author of this book born?
### Composition Task: When was the author of the book "The Great
    Gatsby" born.

### Given Task: Give me the flight schedule from London to
    Edinburgh today.
### Subsequent Task: Which fight has the shortest duration?
### Composition Task: Give me the flight from London to
    Edinburgh with the shortest duration according to the flight
    schedule today.

### Given Task: Retrieve the headlines of the news today from
    BBC.
### Subsequent Task: What is the sentiment of the news
    respectively?
### Composition Task: What is the sentiment of each headline in
    today's news from BBC?

### Given Task: Which team won the World Cup in 2018?
### Subsequent Task: What is the team's captain?
### Composition Task: Who is the captain of the team that won
    the World Cup in 2018.

## Here is the given task, please give your response following
    the above format:
### Given Task: {task}
```

Prompt for Parallel-then-Sequential Composition

```
You are training a model that can take a user's task description
    or query, and available functions as input, and generate a
    sequence of function calls to accomplish the task. Currently,
     you are generating complex tasks for model training. Given a
     task, you need to add a paralle task and a subsequent task
    for this given task to make a more complex task.

The requirements for the parallel task are as follows:
1. The parallel task should be related to the given task, and
    the input should independent of the output of the given task.
2. The parallel task can conduct at the same time as the given
    task, and they can be independent of each other.
3. The output of the given task and the parallel task can be
    used together to conduct a subsequent task.

The requierments for the subsequent task are as follows:
1. The subsequent task should use the output of the given task
    and generate parallel task as input.
2. The subsequent can only be conducted after the given task and
     the parallel task have been completed.
3. The subsequent task, the given task and the parallel task can
     form a new compositional task, and composing them can make a
     more complex multi-step task.

## Examples:
### Given Task: Give me a list of all the pets.
```

```
### Parallel Task: Find available pet food currently in the
   store.
### Subsequent Task: Check if the pet food is suitable for the
   pets in the list.
### Composition Task: Check if the pet food is suitable for the
   pets in the list of all the pets.

### Given Task: When was the author of the book "The Great
   Gatsby" born.
### Parallel Task: Find the publication date of the book "The
   Great Gatsby".
### Subsequent Task: When the book was published, how long had
   it been since the author was born?
### Composition Task: How old was the author of the book "The
   Great Gatsby" when the book was published?

### Given Task: Give me the flight schedule from London to
   Edinburgh today.
### Parallel Task: Find the every hour weather forecast in
   Edinburgh today.
### Subsequent Task: What is the weather condition when the
   first flight arrives?
### Composition Task: I am in London, and I want to know the
   weather condition when the first flight arrives in Edinburgh
   today.

### Given Task: What is the sentiment of each headline in today'
   s news from BBC?
### Parallel Task: Find the sentiment of each headline in today'
   s news from CNN.
### Subsequent Task: Which news source has more positive news
   today?
### Composition Task: Compare the sentiment of each headline in
   today's news from BBC and CNN, and check which news source
   has more positive news.

### Given Task: Who is the captain of the team that won the
   World Cup in 2018?
### Parallel Task: Who is the coach of the team that won the
   World Cup in 2018?
### Subsequent Task: Are the captain and the coach from the same
    country?
### Composition Task: Check if the captain and the coach of the
   team that won the World Cup in 2018 are from the same country
   .

## Here is the given task, please give your response following
   the above format:
### Given Task: {task}
```

The following is the prompt for filtering compositional tasks, and the key idea is to verify the consistency between a compositional task and its atomic sub-tasks.

**Prompt for Filtering Compositional Tasks**

```
You are an expert in task decomposition. Currently, you are
   given a compositional task and its potential task breakdown.
```

```
     Please check if the sub-tasks can be used to complete the
     compositional task.

Compositional task:
{task}

Potential task breakdown:
{sub_tasks}

Please check if the sub-tasks can be used to complete the
    compositional task. You should first give your analysis and
    thinking, and finally give your conclusion (yes or no)
    enclosed in <ans>, for example, <ans>yes</ans> or <ans>no</
    ans>:
```

## A.4 FUNCTION GENERATION

**Prompt for Function Generation**

```
You are training a model that can take a user's task description
     or query, and available functions as input, and generate a
    sequence of function calls to accomplish the task. Currently,
     you are generating the training data for this model.

Given a compositional task and its task breakdown, please
    generate corresponding aviliable functions that can be used
    to accomplish each sub-task, and finally the compositional
    task can be accomplished by calling these functions
    sequentially.

## Requirements for the functions:
1. The functions must possess a succinct, comprehensible name
    and description.
2. The functions should not tailored for a current task, are to
    be used for other future tasks as well, hence the design of
    APIs should be sufficiently generalized.
3. Avoid the recurrence of the task or its components in the
    function description and name, offering a generic perspective
     that can be employed across different contexts.
4. Make every function sufficiently granular and independent,
    avoiding the conflation of multiple tasks within a single
    function and avert creating monolithic APIs.
5. Consistency in terms of parameters and returns from each
    function is critical. For instance, if two functions are
    called sequentially, the output of the first should either
    align with or constitute a part of the input for the second
    function, irrespective of varying parameter terminologies.

## Requirements for the number of functions:
1. One sub-task may need zero, one or multiple functions to
    complete it.
2. If a sub-task is about logic, comparision, set operation or
    calculation, which can be solved by large language models,
    then no function is needed for this sub-task, just leave the
    func_list of this sub-task empty.

## Compositional task:
```

```
{task}

## Task breakdown:
{sub_task}

## Response format:
```json
[
{
    "sub_task": "a sub task from the task breakdown",
    "func_list": [
        {
            "name": "<function name>",
            "description": "<function usage description>",
            "parameters": {
                "<param1>": {
                    "type": "<can be string, number, boolean,
                        object, array, enum and anyOf>",
                    "description": "<param1 description>",
                    ... <more keys if needed>
                },
                ... <more parameters if needed>
            },
            "required": "<array of required parameters, maybe
                not all parameters above are required>"
            "responses": {
                "<res1>" {
                    "type": "<value1 type>",
                    "description": "<value1 description>"
                },
                "<res2>": {
                    "type": "<value2 type>",
                    "description": "<value2 description>"
                }
            }
        },
        {
        ... <more functions if needed>
        }
    ]
}
... <more sub tasks and corresponding functions if needed>
]
```

## Please respond following the format above:
```

## A.5  MULTI-AGENT

**System Prompt for User Agent**

```
Assume that you are a human interacting with an AI assistant.
    You need to engage in a meaningful conversation while always
    remembering to demonstrate human-like behaviour. Avoid
    inquiring if the AI assistant requires assistance, as this
    contradicts your human role. Your main objective is to
    sustain a conversation as a typical user would.
```

```
Currently, your goal is to complete a predefined task, and you
    are seeking the AI assistant for this purpose.

**Task**
{task}

During this conversation, you should take on an active role and
    explore the AI assistant's capability to solve problems \
within the **Task** using a series of function (tool) calls. You
     should adhere to the following guidelines:

1. Your task involves a complex task requiring multiple steps to
     complete. In your initial question to the AI assistant, you
    should provide a detailed explanation of the task, including
    necessary information (such as potential data) that might be
    needed to solve the problem. However, you should withhold
    specific solution steps (e.g., avoid sequential terms like "
    firstly," "secondly") and not dictate which functions (tools)
     the AI should use - that is for the AI to determine.
2. Remember, during this multi-turn dialogue, you are portraying
     the role of a human user. Your questions and responses
    should reflect this human aspect. All your outputs should
    enclose within "<human>" tag, for example, "<human> ... </
    human>".
```

**System Prompt for Assistant Agent**

```
You are simulating the role of an expert in using functions (i.e
    ., tools) to solve users' tasks. You already possess
    knowledge on how to decompose the task into subtasks and
    understand which tools to use for their resolution.

**Subtasks**
{sub_task}

**Available Functions for Subtasks**
{subtask_func}

Please use the tools provided above to answer the question posed
     by "<human>". You must try as much as possible to use these
    tools, instead of directly answering the question using your
    prior knowledge.

Your response must obey the following format:
Observation: Carefully observe the user "<human>"'s question as
    well as the output of the function call (often enclosed
    within the "<func_return>" tag). Be sure to check for any
    errors in previous outputs, as they may not always be
    accurate. Enclose your observation within the "<observation>"
     tag.
Thought: After observing and combining the previously listed
    steps, give detailed and clear thoughts, reasonings, or
    reflections, and according to the plan decide the next step.
    Note: When you believe the task to be complete, you may use '
    final_answer' to provide a detailed summary of the results to
```

```
     give to the user. Enclose your thoughts within the "<thought
     >" tag.
Function Call: Name and arguments of the function call. The
     function name must be same as its name in above function list
     , and the arguments must obey the format required by the
     function. Enclose the function call within the "<func_call>"
     tag. If possible, you can call multiple functions in parallel
     , be sure the functions called parallelly are independent of
     each other.

Example 1 (regular function call):
<observation> User has provided two numbers - 15 and 25. </
     observation>
<thought> Based on user's request, we need to find the greatest
     common divisor of these two numbers. We can use the function
     'find_greatest_common_divisor' to solve this problem. </
     thought>
<func_call>[
{
     "name": "find_greatest_common_divisor",
     "arguments": {"num1": 15, "num2": 25}
}
]</func_call>

Example 2 (parallel function call):
<observation> User wants to know the weather in two cities - New
      York and London. </observation>
<thought> We can use the function 'get_weather' to find the
     weather in New York and London. And the call to this function
      can be done in parallel. </thought>
<func_call>[
{
     "name": "get_weather",
     "arguments": {"city": "New York"}
},
{
     "name": "get_weather",
     "arguments": {"city": "London"}
}
]</func_call>

Example 3 (call final_answer):
<observation> find_greatest_common_divisor returns the result
     "5". </observation>
<thought> The result returned by the function call, along with
     the information collected previously, is sufficient to answer
      the user's question, therefore we now use 'final_answer' to
     provide the user with the answer. </thought>
<function_call>[
{
     "name": "final_answer",
     "arguments": {"final_answer": "5"}
}
]</function_call>

Furthermore, when the user "<human>" raises a question, you need
      to provide a structured plan to solve the question ('
     structured' means that the plan needs to include steps in
```

```
    sequential order, such as Step 1, 2, 3, etc., or logic
    processes that include loops and decision branches). The
    contents of the plan can be placed in the first round
    response's <thought>, and try as much as possible to follow
    this plan in every subsequent function call. However, as
    necessary, you may also modify the relevant plans according
    to the result of the function call.
```

**System Prompt for Tool Agent**

```
You are simulating a computer system with powerful computational
    capabilities and a complete setup. You possess ample
    external prior knowledge, allowing you to run any arbitrary
    function and execute calls to produce results, and you never
    make errors. Give a following function, you should simulate
    the operation of a computer system program as closely as
    possible.

**Function**
{function}

Given a function call, you should execute the function and
    provide the results in JSON format. Your response should
    directly provide the results in JSON format, should not
    contain irrelevant information, and must enclose within "<
    func_return>" tag.

### Example of function return:
<func_call>
{
    "name": "get_weather",
    "arguments": {"city": "New York"}
}

<func_return>
{
    "temperature": "25C",
}
</func_return>
```

## B DATA COLLECTION

### B.1 EXAMPLE OF COLLECTED DATA

**Collected Data Example 1**

```
System:
You are an expert in using functions (i.e., tools) to solve
    users' tasks. The functions available for you to use are
    detailed below:
<tool>[
    {
        "name": "get_current_timestamp",
        "description": "Fetches the current timestamp from the
            device.",
        "parameters": {},
```

```
        "required": []
    },
    {
        "name": "get_humidity_reading",
        "description": "Fetches the current humidity reading
            from a device.",
        "parameters": {
            "device_id": {
                "type": "string",
                "description": "The ID of the device."
            }
        },
        "required": [
            "device_id"
        ]
    },
    {
        "name": "log_data_to_database",
        "description": "Logs data to a server's database.",
        "parameters": {
            "server_id": {
                "type": "string",
                "description": "The ID of the server."
            },
            "data": {
                "type": "object",
                "description": "The data to be logged.",
                "properties": {
                    "message": {
                        "type": "string",
                        "description": "The status update
                            message."
                    },
                    "timestamp": {
                        "type": "string",
                        "description": "The current timestamp."
                    },
                    "temperature": {
                        "type": "number",
                        "description": "The current temperature
                            reading."
                    },
                    "humidity": {
                        "type": "number",
                        "description": "The current humidity
                            reading."
                    }
                },
                "required": [
                    "message",
                    "timestamp",
                    "temperature",
                    "humidity"
                ]
            }
        },
        "required": [
            "server_id",
```

```
                        "data"
                ]
        },
        {
                "name": "send_message_to_server",
                "description": "Sends a message from a device to a
                    server.",
                "parameters": {
                        "device_id": {
                                "type": "string",
                                "description": "The ID of the device sending the
                                    message."
                        },
                        "server_id": {
                                "type": "string",
                                "description": "The ID of the server receiving
                                    the message."
                        },
                        "message": {
                                "type": "string",
                                "description": "The message to be sent."
                        }
                },
                "required": [
                        "device_id",
                        "server_id",
                        "message"
                ]
        },
        {
                "name": "get_temperature_reading",
                "description": "Fetches the current temperature reading
                    from a device.",
                "parameters": {
                        "device_id": {
                                "type": "string",
                                "description": "The ID of the device."
                        }
                },
                "required": [
                        "device_id"
                ]
        }
]</tool>

In your response, you need first provide your observation and
    thought on the user's task, the current situation, and what
    you plan to do next. After your thinking, you can do
    following two things:
**Function Call**: For fountion calling, you need to provide the
     function name and its arguments. The function name must be
    same as its name in above function list, and the arguments
    must obey the format required by the function. Enclose the
    function call within the tag "<call></call>". If possible,
    you can call multiple functions in parallel, be sure the
    functions called parallelly are independent of each other.
**Final Answer**: When you believe the task is complete, you may
     use 'final_answer' to provide a detailed summary of the
```

```
    results to give to the user, enclose the final answer within
    the tag "<final></final>".

User:
Hi, I need to send a status update from DeviceA to ServerB. This
    update should include the current timestamp. Additionally, I
    want to log this status update along with the current
    temperature and humidity readings in the database on ServerB.
    Could you help me with this?

Assistant:
The user wants to send a status update from DeviceA to ServerB
    including the current timestamp, and then log this status
    along with the current temperature and humidity readings on
    ServerB's database. To achieve this task, we will follow
    these steps:
1. Send a status update message from DeviceA to ServerB
    including the device's current timestamp.
    - Fetch the current timestamp from DeviceA.
    - Send the status update message with the timestamp to
        ServerB.
2. Get the current temperature and humidity readings from
    DeviceA.
3. Log the status update message along with the temperature and
    humidity readings on ServerB's database.

We can perform steps 1 and 2 in parallel since they are
    independent of each other. Once we have the required data, we
    will proceed to step 3.
<call>[
    {
        "name": "get_current_timestamp",
        "arguments": {}
    },
    {
        "name": "get_temperature_reading",
        "arguments": {
            "device_id": "DeviceA"
        }
    },
    {
        "name": "get_humidity_reading",
        "arguments": {
            "device_id": "DeviceA"
        }
    }
]</call>

Tool:
[
    {
        "name": "get_current_timestamp",
        "arguments": {},
        "results": {
            "timestamp": "2023-10-03T14:22:00Z"
        }
    },
    {
```

```
        "name": "get_temperature_reading",
        "arguments": {
            "device_id": "DeviceA"
        },
        "results": {
            "temperature": 22.5
        }
    },
    {
        "name": "get_humidity_reading",
        "arguments": {
            "device_id": "DeviceA"
        },
        "results": {
            "humidity": 45.3
        }
    }
]

Assistant:
We have successfully fetched the current timestamp, temperature,
    and humidity readings from DeviceA. The details are as
    follows:
- Timestamp: 2023-10-03T14:22:00Z
- Temperature: 22.5C
- Humidity: 45.3% Now, we need to send a status update message
    from DeviceA to ServerB including the timestamp. After that,
    we will log the status update along with the temperature and
    humidity readings in ServerB's database.
<call>[
    {
        "name": "send_message_to_server",
        "arguments": {
            "device_id": "DeviceA",
            "server_id": "ServerB",
            "message": "Status update at 2023-10-03T14:22:00Z"
        }
    }
]</call>

Tool:
[
    {
        "name": "send_message_to_server",
        "arguments": {
            "device_id": "DeviceA",
            "server_id": "ServerB",
            "message": "Status update at 2023-10-03T14:22:00Z"
        },
        "results": {
            "status": "Message delivered successfully"
        }
    }
]

Assistant:
The status update message was successfully delivered from
    DeviceA to ServerB. The message content was: "Status update
```

```
      at 2023-10-03T14:22:00Z". Now, we need to log this status
      update message along with the temperature and humidity
      readings in ServerB's database. We will use the following
      data:
- Message: "Status update at 2023-10-03T14:22:00Z"
- Timestamp: 2023-10-03T14:22:00Z
- Temperature: 22.5C
- Humidity: 45.3%
<call>[
    {
        "name": "log_data_to_database",
        "arguments": {
            "server_id": "ServerB",
            "data": {
                "message": "Status update at 2023-10-03T14:22:00
                    Z",
                "timestamp": "2023-10-03T14:22:00Z",
                "temperature": 22.5,
                "humidity": 45.3
            }
        }
    }
]</call>

Tool:
[
    {
        "name": "log_data_to_database",
        "arguments": {
            "server_id": "ServerB",
            "data": {
                "message": "Status update at 2023-10-03T14:22:00
                    Z",
                "timestamp": "2023-10-03T14:22:00Z",
                "temperature": 22.5,
                "humidity": 45.3
            }
        },
        "results": {
            "log_status": "Data logged successfully on server
                ServerB"
        }
    }
]

Assistant:
The status update message along with the temperature and
    humidity readings was successfully logged in ServerB's
    database. The task has been completed successfully. A status
    update message was sent from DeviceA to ServerB including the
     current timestamp, and the data was logged on ServerB's
    database along with the current temperature and humidity
    readings.
<final>The status update message including the current timestamp
     was successfully sent from DeviceA to ServerB, and the data
    was logged in ServerB's database along with the temperature
    (22.5C) and humidity (45.3%) readings.</final>
```

## B.2 PARALLEL FUNCTION CALLING

We use different system prompts to guide the behavior of whether to conduct parallel calling or not. We demonstrate both the system prompt allowing parallel calling and the one prohibiting it. The only difference is the instruction at the end of `**Function Call**`. For enabling parallel calling, we use: "If possible, you can call multiple functions in parallel. Be sure the functions called in parallel are independent of each other". For disabling parallel calling, we use: "You should call one function at a time and wait for the response before calling the next function". These instructions help distinguish between turning parallel calling on or off. `{func_list}` is the placeholder for the function definition list in JSON format.

To align LLMs with the correct behavior regarding parallel calling, we implement a 50% probability of turning off parallel calling during trajectory generation for compositional tasks where sub-tasks can be executed in parallel. Since we generate compositional tasks from the bottom up, we inherently know the sub-tasks involved. We fill these trajectories into prompts that prohibit parallel calling. For the other 50% of compositional tasks, where sub-tasks can be executed in parallel, we use system prompts that encourage parallel calling. We also configure trajectories conducted by compositional instructions, where sub-tasks cannot be done in parallel, to use the system prompt that enables parallel calling with a probability of 50%. Finally, the constructed data with the designed prompts can effectively align LLMs to either conduct parallel calling or not, according to the corresponding system prompts. In our work, the accuracy of the parallel calling behavior is not the focus, and we plan to address it in future work.

```
System Prompt Enabling Parallel Calling

You are an expert in using functions (i.e., tools) to solve
    users' tasks. The functions available for you to use are
    detailed below:
<tool>{func_list}</tool>

In your response, you need first provide your observation and
    thought on the user's task, the current situation, and what
    you plan to do next. After your thinking, you can do
    following three things:
**Function Call**: For fountion calling, you need to provide the
     function name and its arguments. The function name must be
    same as its name in above function list, and the arguments
    must obey the format required by the function. Enclose the
    function call within the tag "<call></call>". If possible,
    you can call multiple functions in parallel, be sure the
    functions called in parallel are independent of each other.
**Final Answer**: When you believe the task is complete, you may
     use 'final_answer' to provide a detailed summary of the
    results to give to the user, enclose the final answer within
    the tag "<final></final>".
```

```
System Prompt Disabling Parallel Calling

You are an expert in using functions (i.e., tools) to solve
    users' tasks. The functions available for you to use are
    detailed below:
<tool>{func_list}</tool>

In your response, you need first provide your observation and
    thought on the user's task, the current situation, and what
    you plan to do next. After your thinking, you can do
    following three things:
```

```
**Function Call**: For fountion calling, you need to provide the
    function name and its arguments. The function name must be
    same as its name in above function list, and the arguments
    must obey the format required by the function. Enclose the
    function call within the tag "<call></call>". You should call
    one function at a time, and wait for the response before
    calling the next function.
**Final Answer**: When you believe the task is complete, you may
    use 'final_answer' to provide a detailed summary of the
    results to give to the user, enclose the final answer within
    the tag "<final></final>".
```

### B.3    FUNCTION DISTRIBUTION

We show a sunburst chart of the distribution of generated functions in our proposed
`BUTTONInstruct` in Figure 4, where the inner circle and outer circle represent the first and sec-
ond words in a function name, respectively. It shows the diversity of our synthesized data, and the
distribution of these functions is also consistent with our daily tasks.

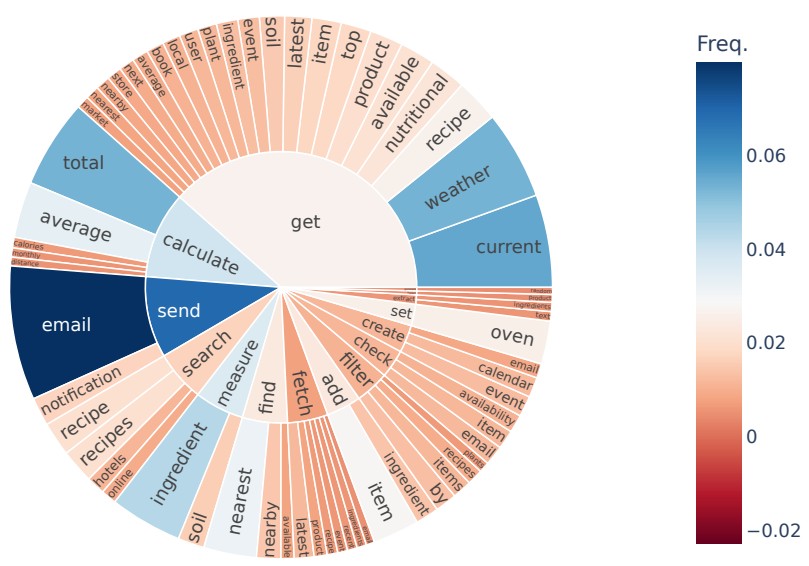

Figure 4: Distribution of the generated functions in `BUTTONInstruct`.

## C    EXPERIMENTS

### C.1    EVALUATION METRICS

For metric calculation for GTA and Tool-Query, we follow the methodologies outlined in their orig-
inal papers with slight modifications. Specifically, during the end-to-end evaluation mode in GTA,
the MathOCR functionality is implemented using the Mathpix API (https://mathpix.com/). How-
ever, since this API requires a subscription and we needed to conduct numerous pilot experiments
and analyses, we did not evaluate questions on end-to-end mode that would invoke this API. The

number of test samples excluding these questions is 209. For Tool-Query, we refined the metric calculation by comparing the final answer with the ground truth. The original implementation was based on exact match; however, such strategies can overlook many successful answers. For example, consider the question "Which paper has received more citations: 'Stability and Risk Bounds of Iterative Hard Thresholding' or 'Compressive Wideband Spectrum Sensing and Signal Recovery With Unknown Multipath Channels'?", models may finished with "The paper 'Stability and Risk Bounds of Iterative Hard Thresholding' has received more citations (5) compared to 'Compressive Wideband Spectrum Sensing and Signal Recovery With Unknown Multipath Channels' (2)". However, the ground truth answer is labeled as "Stability and Risk Bounds of Iterative Hard Thresholding", and exact match strategies may incorrectly label this answer as unsuccessful. Thus, we refined the metric strategy to check if the final answer is present in the model output. Note that all these modifications are applied to all models, including baselines, ensuring that the comparison between baselines and models is fair.

## C.2 IMPLEMENTATION

All instruction-tuning training is performed on $4 \times 8$ NVIDIA H800 GPUs, using the training framework based on HuggingFace Transformers (Wolf et al., 2019). We use the corresponding instruction format for Llama and Qwen models. To enhance training efficiency, we pack short instances into longer ones and apply flash attention. During model training, we optimize the loss only on the response content from assistant roles. We use a learning rate of 2e-5 with cosine decay and a batch size of 64 for all models. For Llama3-8B and Qwen2-7B, we train for five epochs, and for Llama3-70B and Qwen2-72B, we train for two epochs.

## C.3 PROMPTS FOR ABLATION STUDY

Here is the monolithic prompt for the 'w/o Bottom-Up Setting' in the ablation study. The placeholders {scen} represents the given scenario. This prompt is used to generate compositional tasks directly, without our proposed bottom-up process.

---

**Monolithic Task Construction Prompt for the w/o Bottom-Up Setting**

```
You are training a model that can take a user's task description
    or query, and available functions as input, and generate a
   sequence of function calls to accomplish the task. Currently,
    you are generating the training data for this model. Given a
    general life scenario as the context, please first generate
   a task.

## Requirements for each task:
1. The task should be accomplishable by calling multiple
    functions with multiple and no more than 7 steps (i.e., turns
    ).
2. If you mention some information, criteria or constraints in
    the task, please give the details of these information,
    criteria or constraints. Do not assume the model has access
    to your personal information or prior knowledge, and it does
    not have chance to ask you for clarification.
3, Please give enough details and make the task description as
    specific as possible, so the model can make deterministic
    function calls with deterministic arguments. Do not include
    any ambiguous or vague information.
4. Do not mention specific tools or functions in the task
    description, and do not propose solutions, hints, or project
    outcomes.
5. Limit the task description to 30 words, and avoid using
    adjectives and ambiguous words.
```

```
## Given scenario:
{scen}

## Please respond a task directly following the requirements
    above in one line:
```

Here are the monolithic prompts for the 'w/o Top-Down Setting' in the ablation study. The two prompts are used for tasks with or without parallel function calling, respectively. The placeholders {task}, {sub_task}, and {subtask_func} represent a specific generated task, its sub-tasks, and the corresponding generated functions for the sub-tasks.

---

**Monolithic Trajectory Generation Prompt for the w/o Top-Down Setting (Non-parallel)**

```
You are labeling data for training an AI assistant that can
    solve a complex compositional task by using tools in a multi-
    turn manner.  Given a complex compositional task, its
    potential subtasks and the available tools (i.e., functions)
    to solve these subtasks, you should generate synthetic data
    about the trajectory of solving the task by using tools in a
    multi-turn manner.

**Task**
{task}

**Subtasks**
{sub_task}

**Available Functions for Sub-tasks**
{subtask_func}

During trajectory generation, you should simulate three roles:
1. human: ask questions to the assistant
2. assistant: answer the questions of human by leveraging the
    tools
3. tool: execute the functions and return the results to the
    assistant

The requirements for the human are:
The human need give the initial question to the AI assistant
    based on the given task, and should provide a detailed
    explanation of the task, including necessary information (
    such as potential data) that might be needed to solve the
    problem. However, you should withhold specific solution steps
     (e.g., avoid sequential terms like "firstly," "secondly")
    and not dictate which functions (tools) the AI should use -
    that is for the AI to determine.

The requirements for the assistant are:
1. The assistant should use the tools provided above to answer
    the question posed by human. It must try as much as possible
    to use these tools, instead of directly answering the
    question using your prior knowledge.
2. The format must obey the following format:
   a) Observation. Carefully observe the human's question as
       well as the output of the tool call. Be sure to check for
        any errors in previous outputs, as they may not always
       be accurate. Enclose your observation within the "<
       observation>" tag.
```

```
        b) Thought. Give detailed and clear thoughts, reasoning, or
           reflections, and according to the plan decide the next
           step. Enclose your thoughts within the "<thought>" tag.
        c) Function call. Give the name and arguments of the
           function call. The function name must be same as its name
            in above function list, and the arguments must obey the
           format required by the function. The function call should
            enclosed within the "<func_call>" tag. Note: When you
           believe the task to be complete, you may use '
           final_answer' to provide a detailed summary of the
           results to give to the user.

The requirements for the tool are:
1. You should simulate the operation of a computer system
   program as closely as possible. Your response should directly
    provide the results in JSON format, should not contain
   irrelevant information.
2. You should also provide the response in the format of JSON
   including name, arguments and results.

### Example:
<role>Human</role>
Hi, I want to know when was the author of "The Great Gatsby" was
    born.
<role>Assistant</role>
<observation>The user has provided a book name "The Great Gatsby
   " and wants to know the author's birthdate.</observation>
<thought>We can use the function 'get_book_author' to find the
   author of the book and the 'get_author_info' to find the
   birthdate of the author.</thought>
<func_call>[
{
    "name": "get_book_author",
    "arguments": {"book_name": "The Great Gatsby"}
}
]</func_call>
<role>Tool</role>
[{
    "name": "get_book_author",
    "arguments": {"book_name": "The Great Gatsby"},
    "results": {"name": "F. Scott Fitzgerald"}
}]
<role>Assistant</role>
<observation>The function 'get_book_author' returns the author
   of the book "The Great Gatsby" is F. Scott Fitzgerald.</
   observation>
<thought>Now we can use the function 'get_author_info' to find
   the birthdate of the author.</thought>
<func_call>[
{
    "name": "get_author_info",
    "arguments": {{"name": "F. Scott Fitzgerald"}}
}
]</func_call>
<role>Tool</role>
[{
    "name": "get_author_info",
    "arguments": {"name": "F. Scott Fitzgerald"}
```

```
    "results": {"birthdate": "September 24, 1896"}
}]
<role>Assistant</role>
<observation>The function 'get_author_info' returns the
    birthdate of the author "F. Scott Fitzgerald" is September
    24, 1896.</observation>
<thought>The result returned by the function call, along with
    the information collected previously, is sufficient to answer
     the user's question, therefore we now use 'final_answer' to
    provide the user with the answer.</thought>
<func_call>[
{
    "name": "final_answer",
    "arguments": {"final_answer": "F. Scott Fitzgerald was born
        on September 24, 1896."}
}
]</func_call>

You need only generate the trajectory in the above format,
    without any other explanation or comments.
```

---

**Monolithic Trajectory Generation Prompt for the w/o Top-Down Setting (Parallel)**

```
You are labeling data for training an AI assistant that can
    solve a complex compositional task by using tools in a multi-
    turn manner. Given a complex compositional task, its
    potential subtasks and the available tools (i.e., functions)
    to solve these subtasks, you should generate synthetic data
    about the trajectory of solving the task by using tools in a
    multi-turn manner.

**Task**
{task}

**Subtasks**
{sub_task}

**Available Functions for Subtasks**
{subtask_func}

During trajectory generation, you should simulate three roles:
1. human: ask questions to the assistant
2. assistant: answer the questions of human by leveraging the
    tools
3. tool: execute the functions and return the results to the
    assistant

The requirements for the human are:
The human need give the initial question to the AI assistant
    based on the given task, and should provide a detailed
    explanation of the task, including necessary information (
    such as potential data) that might be needed to solve the
    problem. However, you should withhold specific solution steps
     (e.g., avoid sequential terms like "firstly," "secondly")
    and not dictate which functions (tools) the AI should use –
    that is for the AI to determine.
```

```
The requirements for the assistant are:
1. The assistant should use the tools provided above to answer
   the question posed by human. It must try as much as possible
   to use these tools, instead of directly answering the
   question using your prior knowledge.
2. The format must obey the following format:
    - a) Observation. Carefully observe the human's question as
         well as the output of the tool call. Be sure to check for
         any errors in previous outputs, as they may not always
         be accurate. Enclose your observation within the "<
         observation>" tag.
    - b) Thought. Give detailed and clear thoughts, reasonings,
         or reflections, and according to the plan decide the next
         step. Enclose your thoughts within the "<thought>" tag.
    - c) Function call. Give the name and arguments of the
         function call. The function name must be same as its name
         in above function list, and the arguments must obey the
         format required by the function. The function call should
         enclosed within the "<func_call>" tag. If possible, you
         can call multiple functions in parallel, be sure the
         functions called parallelly are independent of each other
         . Note: When you believe the task to be complete, you may
         use 'final_answer' to provide a detailed summary of the
         results to give to the user.

The requirements for the tool are:
1. You should simulate the operation of a computer system
   program as closely as possible. Your response should directly
   provide the results in JSON format, should not contain
   irrelevant information.
2. You should also provide the response in the format of JSON
   including name, arguments and results.

### Example 1:
<role>Human</role>
Hi, I want to know when was the author of "The Great Gatsby" was
    born.
<role>Assistant</role>
<observation>The user has provided a book name "The Great Gatsby
    " and wants to know the author's birthdate.</observation>
<thought>We can use the function 'get_book_author' to find the
    author of the book and the 'get_author_info' to find the
    birthdate of the author.</thought>
<func_call>[
{
    "name": "get_book_author",
    "arguments": {"book_name": "The Great Gatsby"}
}
]</func_call>
<role>Tool</role>
[{
    "name": "get_book_author",
    "arguments": {"book_name": "The Great Gatsby"},
    "results": {"name": "F. Scott Fitzgerald"}
}]
<role>Assistant</role>
```

```
<observation>The function 'get_book_author' returns the author
    of the book "The Great Gatsby" is F. Scott Fitzgerald.</
    observation>
<thought>Now we can use the function 'get_author_info' to find
    the birthdate of the author.</thought>
<func_call>[
{
    "name": "get_author_info",
    "arguments": {"name": "F. Scott Fitzgerald"}
}
]</func_call>
<role>Tool</role>
[{
    "name": "get_author_info",
    "arguments": {"name": "F. Scott Fitzgerald"}
    "results": {"birthdate": "September 24, 1896"}
}]
<role>Assistant</role>
<observation>The function 'get_author_info' returns the
    birthdate of the author "F. Scott Fitzgerald" is September
    24, 1896.</observation>
<thought>The result returned by the function call, along with
    the information collected previously, \
is sufficient to answer the user's question, therefore we now
    use 'final_answer' to provide the user with the answer.</
    thought>
<func_call>[
{
    "name": "final_answer",
    "arguments": {{"final_answer": "F. Scott Fitzgerald was born
        on September 24, 1896."}}
}
]</func_call>

### Example 2:
<role>Human</role>
What is the age when the author of "The Great Gatsby" published
    the book.
<role>Assistant</role>
<observation>The user mention the book "The Great Gatsby" and
    wants to know the how old the author was when he published
    the book.</observation>
<thought>We can use the function 'get_book_author' to find the
    author of the book and the 'get_book_publish_date' to find
    the publish date of the book, such two function can be used
    in parallel.</thought>
<func_call>[
{
    "name": "get_book_author",
    "arguments": {"book_name": "The Great Gatsby"}
},
{
    "name": "get_book_publish_date",
    "arguments": {"book_name": "The Great Gatsby"}
}
]</func_call>
<role>Tool</role>
[{
```

```
    "name": "get_book_author",
    "arguments": {"book_name": "The Great Gatsby"},
    "results": {"name": "F. Scott Fitzgerald"}
},
{
    "name": "get_book_publish_date",
    "arguments": {"book_name": "The Great Gatsby"},
    "results": {"publish_date": "April 10, 1925"}
}]
<role>Assistant</role>
<observation>The function 'get_book_author' returns the author
    of the book "The Great Gatsby" is F. Scott Fitzgerald, and
    the function 'get_book_publish_date' returns the publish date
     of the book "The Great Gatsby" is April 10, 1925.</
    observation>
<thought>To determine the age of the author when the book was
    published, we need to calculate the difference between the
    publish date and the author's birthdate, so we need to use
    the function 'get_author_info' to get the birthdate of the
    author.</thought>
<func_call>[
{
    "name": "get_author_info",
    "arguments": {"name": "F. Scott Fitzgerald"}
}
]</func_call>
<role>Tool</role>
[{
    "name": "get_author_info",
    "arguments": {"name": "F. Scott Fitzgerald"},
    "results": {"birthdate": "September 24, 1896"}
}]
<role>Assistant</role>
<observation>The function 'get_author_info' returns the
    birthdate of the author "F. Scott Fitzgerald" is September
    24, 1896.</observation>
<thought>Previously, we have got the publish date of the book "
    The Great Gatsby" is April 10, 1925, so we can infer that the
     author was 28 years old when he published the book.</thought
    >
<func_call>[
{
    "name": "final_answer",
    "arguments": {"final_answer": "F. Scott Fitzgerald was 28
        years old when he published "The Great Gatsby"."}
}
]</func_call>

You need only generate the trajectory in the above format,
    without any other explanation or comments.
```

## C.4 CASES ON PARALLEL CALLING

We present two cases of conducting Llama3-70B-BUTTON on Tool-Query. In both cases, the functions authorEdgeCheck and paperNodeCheck are called in parallel where appropriate. This parallel calling effectively reduces the number of turns during the multi-turn function calling procedure. For example, in case 2, with parallel calling, the language model only requires 6 steps

(including providing the final answer) to complete the task. Without parallel calling, 2 of these steps would expand to 4 steps each, resulting in a total of 12 steps.

```
Parallel Calling Case 1

User: Who has the most collaborations with Lazhar Labiod in the
    DBLP citation network?

Assistant: To answer this question, we need to follow these
    steps:
1. Load the AuthorNet.
2. List the first-order neighbors (collaborators) of Lazhar
    Labiod.
3. Determine which collaborator has the most collaborations with
     Lazhar Labiod.

Let's start by loading the AuthorNet.
<call>[
    {
        "name": "loadAuthorNet",
        "arguments": {}
    }
]</call>

Tool: AuthorNet is loaded.

Assistant: AuthorNet is loaded successfully. Next, we need to
    list the first-order neighbors (collaborators) of Lazhar
    Labiod.
<call>[
    {
        "name": "neighbourCheck",
        "arguments": {
            "graph": "AuthorNet",
            "node": "Lazhar Labiod"
        }
    }
]</call>

Tool: ['Mohamed Nadif', 'Chakib Fettal']

Assistant: The collaborators of Lazhar Labiod are Mohamed Nadif
    and Chakib Fettal. Now, we need to determine which of these
    collaborators has the most collaborations with Lazhar Labiod.
     We can do this by checking the number of co-authored papers
    between Lazhar Labiod and each of his collaborators.
<call>[
    {
        "name": "authorEdgeCheck",
        "arguments": {
            "node1": "Lazhar Labiod",
            "node2": "Mohamed Nadif"
        }
    },
    {
        "name": "authorEdgeCheck",
        "arguments": {
            "node1": "Lazhar Labiod",
```

```
                "node2": "Chakib Fettal"
            }
        }
]</call>

Tool: [{'collaborative_papers': ['TensorClus: A python library
    for tensor (Co)-clustering', 'Efficient Graph Convolution for
     Joint Node Representation Learning and Clustering']}", "{'
    collaborative_papers': ['Efficient Graph Convolution for
    Joint Node Representation Learning and Clustering']}]

Assistant: Lazhar Labiod has co-authored 2 papers with Mohamed
    Nadif and 1 paper with Chakib Fettal. Based on the number of
    co-authored papers, Mohamed Nadif has the most collaborations
     with Lazhar Labiod. Now, I will provide the final answer to
    the user.
<final>Mohamed Nadif has the most collaborations with Lazhar
    Labiod, with a total of 2 co-authored papers.</final>
```

### Parallel Calling Case 2

```
User: How many citations do papers co-authored by Florian
    Kirchbuchner and Fadi Boutros have in the DBLP citation
    network? Please give me a numerical value as an answer.

Assistant: To answer this question, we need to follow these
     steps:
1. Load the AuthorNet.
2. Check the collaboration relationship between Florian
    Kirchbuchner and Fadi Boutros.
3. Count the total number of citations of the papers they co-
    authored.

Let's start by loading the AuthorNet.
<call>[
    {
        "name": "loadAuthorNet",
        "arguments": {}
    }
]</call>

Tool: AuthorNet is loaded.

Assistant: AuthorNet is loaded successfully. Next, we need to
    check the collaboration relationship between Florian
    Kirchbuchner and Fadi Boutros to get the list of papers they
    co-authored.
<call>[
    {
        "name": "authorEdgeCheck",
        "arguments": {
            "node1": "Florian Kirchbuchner",
            "node2": "Fadi Boutros"
        }
    }
]</call>
```

```
Tool: {'collaborative_papers': ['PocketNet: Extreme Lightweight
    Face Recognition Network Using Neural Architecture Search and
     Multistep Knowledge Distillation', 'ElasticFace: Elastic
    Margin Loss for Deep Face Recognition.', 'Template-Driven
    Knowledge Distillation for Compact and Accurate Periocular
    Biometrics Deep-Learning Models', 'Self-restrained triplet
    loss for accurate masked face recognition']}

Assistant: The papers co-authored by Florian Kirchbuchner and
    Fadi Boutros are:
1. PocketNet: Extreme Lightweight Face Recognition Network Using
     Neural Architecture Search and Multistep Knowledge
    Distillation
2. ElasticFace: Elastic Margin Loss for Deep Face Recognition.
3. Template-Driven Knowledge Distillation for Compact and
    Accurate Periocular Biometrics Deep-Learning Models
4. Self-restrained triplet loss for accurate masked face
    recognition

We need to find the total number of citations for these papers.
    To find the total number of citations for these papers, we
    will check the citation count for each paper individually and
     then sum them up.
<call>[
    {
        "name": "paperNodeCheck",
        "arguments": {
            "node": "PocketNet: Extreme Lightweight Face
                Recognition Network Using Neural Architecture
                Search and Multistep Knowledge Distillation"
        }
    },
    {
        "name": "paperNodeCheck",
        "arguments": {
            "node": "ElasticFace: Elastic Margin Loss for Deep
                Face Recognition."
        }
    },
    {
        "name": "paperNodeCheck",
        "arguments": {
            "node": "Template-Driven Knowledge Distillation for
                Compact and Accurate Periocular Biometrics Deep-
                Learning Models"
        }
    },
    {
        "name": "paperNodeCheck",
        "arguments": {
            "node": "Self-restrained triplet loss for accurate
                masked face recognition"
        }
    }
]</call>
```

```
Tool: ['Please load the PaperNet first.', 'Please load the
    PaperNet first.', 'Please load the PaperNet first.', 'Please
    load the PaperNet first.']

Assistant: The system requires PaperNet to be loaded before
    checking the details of individual papers. We need to load
    PaperNet first before proceeding to check the citation counts
     of the papers.
<call>[
    {
        "name": "loadPaperNet",
        "arguments": {}
    }
]</call>

Tool: PaperNet is loaded.

Assistant: PaperNet is loaded successfully. Now that PaperNet is
     loaded, we can proceed to check the citation counts for each
     of the papers co-authored by Florian Kirchbuchner and Fadi
    Boutros.
<call>[
    {
        "name": "paperNodeCheck",
        "arguments": {
            "node": "PocketNet: Extreme Lightweight Face
                Recognition Network Using Neural Architecture
                Search and Multistep Knowledge Distillation"
        }
    },
    {
        "name": "paperNodeCheck",
        "arguments": {
            "node": "ElasticFace: Elastic Margin Loss for Deep
                Face Recognition."
        }
    },
    {
        "name": "paperNodeCheck",
        "arguments": {
            "node": "Template-Driven Knowledge Distillation for
                Compact and Accurate Periocular Biometrics Deep-
                Learning Models"
        }
    },
    {
        "name": "paperNodeCheck",
        "arguments": {
            "node": "Self-restrained triplet loss for accurate
                masked face recognition"
        }
    }
]</call>

Tool: {'authors': ['Fadi Boutros', 'Naser Damer', 'Florian
    Kirchbuchner', 'Arjan Kuijper'], 'year': 2022, 'venue': 'IEEE
     ACCESS', 'n_citation': 0, 'keywords': ['Computer
    architecture', 'Microprocessors', 'Training', 'Task analysis
```

```
      ', 'Network architecture', 'Computational modeling', '
      Architecture', 'Computer vision', 'deep learning', 'face
      recognition'], 'doc_type': 'Journal'}", "{'authors': ['Fadi
      Boutros', 'Naser Damer', 'Florian Kirchbuchner', 'Arjan
      Kuijper'], 'year': 2022, 'venue': 'IEEE Conference on
      Computer Vision and Pattern Recognition', 'n_citation': 43, '
      keywords': [], 'doc_type': 'Conference'}", "{'authors': ['
      Fadi Boutros', 'Naser Damer', 'Florian Kirchbuchner', 'Arjan
      Kuijper'], 'year': 2022, 'venue': 'SENSORS', 'n_citation': 0,
       'keywords': ['biometrics', 'knowledge distillation', '
      periocular verification'], 'doc_type': 'Journal'}", "{'
      authors': ['Fadi Boutros', 'Naser Damer', 'Florian
      Kirchbuchner', 'Arjan Kuijper'], 'year': 2022, 'venue': '
      Pattern Recognition', 'n_citation': 8, 'keywords': ['COVID
      -19', 'Biometric recognition', 'Identity verification', '
      Masked face recognition'], 'doc_type': 'Journal'}

Assistant: The citation counts for the papers co-authored by
      Florian Kirchbuchner and Fadi Boutros are as follows:
1. PocketNet: Extreme Lightweight Face Recognition Network Using
       Neural Architecture Search and Multistep Knowledge
      Distillation - 0 citations
2. ElasticFace: Elastic Margin Loss for Deep Face Recognition. -
       43 citations
3. Template-Driven Knowledge Distillation for Compact and
      Accurate Periocular Biometrics Deep-Learning Models - 0
      citations
4. Self-restrained triplet loss for accurate masked face
      recognition - 8 citations

The total number of citations is the sum of these counts. The
      total number of citations for the papers co-authored by
      Florian Kirchbuchner and Fadi Boutros is \(0 + 43 + 0 + 8 =
      51\). I will now provide the final answer.
<final>The total number of citations for the papers co-authored
      by Florian Kirchbuchner and Fadi Boutros is 51.</final>
```

