# OpenReview forum: "Facilitating Multi-turn Function Calling for LLMs via Compositional Instruction Tuning"
_ICLR.cc/2025/Conference — ICLR 2025 Poster_

### Official Review · Reviewer_5MrE · 2024-10-28

**Soundness:** 3
**Presentation:** 3
**Contribution:** 2
**Rating:** 5
**Confidence:** 3

**Summary:**

This paper introduces BUTTON, a generation pipeline to improve multi-turn function calling for LLMs through compositional instruction tuning, to handle complex real-world tasks requiring sequential or parallel function calls.  BUTTON uses a combination of bottom-up task creation and top-down trajectory generation in a multi-agent environment. This pipeline also produces BUTTONInstruct, a dataset with 8,000 multi-turn function call trajectories, and demonstrates improved model performance across various benchmarks.

**Strengths:**

1. This paper offers a scalable solution for creating synthetic multi-turn datasets, enhancing LLM capabilities in function calling. This is an important feature for AI agents in many applications.

2. The BUTTON pipeline is innovative in combining both bottom-up and top-down data generation processes to address the need for getting complex, multi-turn interactions data  from real-world.

3. The performance seems good. After finetuned with the generated dataset, all models get improved in terms of accuracy.

**Weaknesses:**

1. While the BUTTON pipeline effectively generates compositional tasks and interaction trajectories, the paper lacks a rigorous quality control process. For example, it does not sufficiently address how to ensure that two atomic tasks are logically compatible for creating a meaningful complex task. Furthermore, the paper mentions filtering compositional tasks by “checking whether each one can be completed by its atomic sub-tasks.” However, it remains unclear whether this condition alone is adequate to ensure that the composed tasks are realistic and applicable to real-world scenarios.

2. The generated function calls are conceptual, lacking actual implementations, which may limit their practical usability. Without concrete implementations, there is a risk that these generated function calls might not be feasible in real-world applications. Additionally, it’s unclear whether the pipeline checks for duplicated function calls or if previously generated functions can be reused in later processes, both of which could enhance efficiency and coherence in the generated dataset.

**Questions:**

1. The pipeline effectively generates parallel-then-sequential compositions, but how does BUTTON determine which tasks are best suited for parallel execution? Are there specific criteria or heuristics that guide the system in identifying parallelizable tasks?

2. What is the seed data to generate the atomic tasks?

3. How many atomic tasks can be used to generate a composed task at most? Does it affect the data complexity and fine-tuned performance?

---

> ### Author Response · Authors · 2024-11-22
>
> We sincerely appreciate the reviewer's efforts and insightful comments to improve our manuscript. Below, we address the concerns (W) and questions (Q) raised.
>
> **[W1] While the BUTTON pipeline effectively generates compositional tasks and interaction trajectories, the paper lacks a rigorous quality control process.**
>
> In the context of data synthesis, data quality can be ensured through either generation strategies or post-generation filtering strategies. We believe that finding a universally applicable filtering strategy is unrealistic, especially considering our scenarios where we need to generate complex compositional tasks, functions and corresponding multi-turn function calling trajectories.
>
> Thus, our approach focuses on controlling quality at the source of data generation, which is reflected in the elaborate design of different phases in our framework. This is also one of the motivations of our work. Specifically,
> - In the bottom-up phase, we start with atomic tasks rather than generating complex tasks directly, which helps construct high-quality tasks and corresponding functions.
> - In the top-down phase, we use a multi-agent approach to simulate the calling trajectory, avoiding the use of a single monolithic prompt, which aids in generating high-quality trajectories.
>
> Although 'quality' is not easily quantified, our empirical results and ablation studies demonstrate the effectiveness of our approach. In summary, we believe that our approach effectively controls data quality at the source of generation, addressing the problem at its source rather than relying solely on post-generation filtering. The latter may need complex heuristic rules or entail large labor costs, which is infeasible in our complex scenarios.
>
> **[W2.1] The generated function calls are conceptual, lacking actual implementations, which may limit their practical usability.**
>
> While we use generated functions during the data creation process, it's crucial to highlight that our test datasets consist of real-world functions and manually annotated queries. These real-world functions include actual implementations that can be executed with actual environments. Our experiments demonstrate that models fine-tuned with our synthetic data generalize effectively to these real-world test sets, highlighting the practical usability of our generated data.
>
> Moreover, as described in the "Function Generation" phase in Section 2.1 (Section 3.1 in new version), there are several advantages to using generated functions. First, by creating compositional tasks before generating functions, we ensure compatibility between tasks and functions. The function-agnostic task generation avoids constraints that could arise if functions were generated first. For instance, selecting functions first and then creating tasks might limit the tasks' scope and realism. Second, relying solely on existing functions with actual implementations would confine tasks and functions to a predefined set, reducing diversity. In contrast, our method promotes flexibility and diversity, enhancing the model's adaptability across a broader range of scenarios. Additionally, our datasets can be easily scaled, providing further benefits.
>
> Thus, we believe that using generated functions during the data synthesis process is a practical and effective approach, and won't limit the practical usability of the generated data, which can be verified by our empirical results.
>
> **[W2.2] It’s unclear whether the pipeline checks for duplicated function calls or if previously generated functions can be reused in later processes**
>
> For each specific compositional task, our pipeline includes a check for duplicated generated functions to ensure that all functions within that task are unique. This helps maintain the integrity and realism of each task's function set.
>
> However, we do not check for duplication across different tasks. Each task has its own unique set of functions, and we do not anticipate reusing generated functions across tasks.
> Our framework is focused on enhancing generalization in function-calling abilities rather than memorizing specific functions, so reusing functions across tasks does not contribute to this goal.
>
> Specifically, available functions for a task are treated as "variables" inserted into the system prompt for the task, as demonstrated in Appendix B.1. This helps LLMs understand and leverage the given functions rather than memorizing all functions during training. This setting aligns with standard practices in function calling for LLMs [1,2].

---

> ### Author Response · Authors · 2024-11-22
>
> **[Q1] How does BUTTON determine which tasks are best suited for parallel execution?**
>
> This is also one of the question and challenge when we began synthesizing data for function calling in LLMs, and it motivated the design of our bottom-up phase. *Instead of trying to determine which tasks are inherently suited for parallel execution, we focus on generating tasks that require parallel execution*, which is exactly the bottom-up phase in our framework.
>
> This strategy is highly beneficial. By knowing the detailed sub-tasks of a compositional task, we can optimize the data synthesis pipeline to generate functions and trajectories that are more compatible with the task, since these processes are aware of and can be guided by the sub-tasks (also mentioned in line 84-88 in our paper).
>
> Importantly, these sub-task details are used solely for the data synthesis pipeline and are not included in the final data. Our goal is to enable LLMs to independently understand and handle compositional tasks.
>
> **[Q2] What is the seed data to generate the atomic tasks?**
>
> In our work, the seed data for scenario extraction is derived from the glaive-function-calling-v2 [3] and ToolLLama datasets [4].
> We have provided this information in Appendix A.1.
>
> **[Q3] How many atomic tasks can be used to generate a composed task at most? Does it affect the data complexity and fine-tuned performance?**
>
> In Section 2.1 (Section 3.1 in new version), under "Compositional Task Construction," we explain our use of heuristic strategies to create compositional tasks. Each compositional task originates from a single atomic task. For Sequential Composition, we generate a subsequent task. In Parallel-then-Sequential Composition, we first create a task that can be executed in parallel with the atomic task, followed by a subsequent task based on the outcomes of the initial two tasks. Thus 2 or 3 atomic tasks are used to generate a composed task.
>
> While it's possible to use an arbitrary number of atomic tasks to create complex compositional tasks by repeatedly applying these strategies, our pilot experiments indicate that extending them too far can negatively impact performance. We believe this is because the current BUTTON framework already produces complex and diverse tasks requiring multi-turn function calls. Repeated application often leads to overly complex tasks with limited practical value.
>
> Despite not extending the simple heuristic, our collected data remains diverse. As mentioned in Section 2.3 (Section 3.3 in new version), "Data Diversity", by analyzing the collected data, we note that the total number of assistant turns ranges from 2 to 7, and the number of function calls can be as high as 6. This indicates that our compositional tasks are not confined to just 2 or 3 sub-tasks. This is because the total number of sub-tasks is not solely determined by the heuristic strategy. During function generation, multiple functions may be created for a single atomic task, allowing for the expansion of a single sub-task. An concrete example can be found in Appendix B.1.
>
> Moreover, as demonstrated in our experiments, the LLMs fine-tuned with our generated data effectively handle tasks with different length of sub-tasks. For instance, the labeled number of turns of queries in GTA are ranged from 2 to 8 (Figure 3(a) in [1]) and the average number of turns is 5 in Tool-Query (Table 3 in [2]). In our evaluation on Tool-Query, tasks needing fewer or more than 4 turns were categorized as easy and hard, respectively (Table 3). Our model excelled in both categories, indicating its capability to handle tasks with different length of sub-tasks.
>
> In summary, the number of atomic tasks used to generate a composed task 2 or 3, however the complexity of the final compositional tasks are not limited by this number. The experiment also show the effectiveness of our approach on multi-turn function calling tasks with different length.
>
> ---
>
> > [1] GTA: A Benchmark for General Tool Agents
> [2] AgentBoard: An Analytical Evaluation Board of Multi-turn LLM Agents
> [3] https://huggingface.co/datasets/glaiveai/glaive-function-calling-v2
> [4] ToolLLM: Facilitating Large Language Models to Master 16000+ Real-world APIs

---

> > ### Comment · Reviewer_5MrE · 2024-11-27
> > **Rebuttal Response**
> >
> > Thanks to the authors for the rebuttal. After careful consideration, I have decided to maintain my original score.

---

> > > ### Author Response · Authors · 2024-12-03
> > >
> > > Dear Reviewer 5MrE:
> > >
> > > Thank you very much for your time and effort in reviewing our manuscript. We appreciate your careful consideration of our rebuttal and the feedback you have provided. We understand and respect your decision.

---

### Official Review · Reviewer_piq6 · 2024-11-05

**Soundness:** 4
**Presentation:** 4
**Contribution:** 4
**Rating:** 8
**Confidence:** 5

**Summary:**

The authors found that when large language models are called external functions, there is another important but neglected problem: how should the model plan and use these functions when dealing with complex tasks that require multiple rounds of dialogue?

To solve this problem, they proposed a method called **BUTTON**. This method is divided into two steps:

- The first step is to build training data from the bottom up. They first designed some simple basic tasks and then combined them into more complex ones. Then, they developed corresponding functional interfaces for these tasks.

- The second step is to generate dialogues from the top down. They built a multi-agent environment to allow simulated users, assistants, and tools to interact and generate data for multiple rounds of dialogue.

In this way, they generated 8,000 high-quality training data. Experimental results show that models trained with this data perform better when dealing with complex tasks.

This research is efficient because tasks in real life often require multiple steps to complete, and models must learn to plan what functions to use and when to use them.

**Strengths:**

**Originality**

Currently, large prediction models are very popular, and most researchers are studying how to make the model give higher-quality human-like text in dialogue. The authors found two difficulties: 1. The instructions of multi-round dialogues are too complex to be recognized by the model, and 2. There are problems with the compatibility of instructions and functions.

**Quality**

The author proposed a new method, the BUTTON method, which has a clear technical route and includes two stages: 1) bottom-up instruction construction and 2) top-down trajectory generation. Two professional benchmarks (GTA and Tool-Query) were used to evaluate performance during the experiment. The scale of 8,000 data points is medium-sized in current similar studies, but it includes multiple roles such as system, user, assistant, tool, etc., which enriches the interaction scenario. The experimental results compare several currently very popular large language models, which is convincing.

**Clarity**

The paragraphs of the article are very clear and easy to understand. The technical terms are also explained.

**Significance**

This study solves the data acquisition problem through synthetic data and function call mechanism, which not only reduces the development cost but also expands the practical scenarios of LLM, which is of great significance in promoting the implementation of AI technology in practical applications.

**Weaknesses:**

The sample size of the benchmark GTA is small, with only 229 queries, and the benchmark Tool-Query has only 60 tasks, which may not fully cover the actual application scenarios. At the same time, the benchmark may have a subjective bias because the manually written queries may carry the subjective judgment of the writer, and the difficulty classification criteria may not be objective enough. Suggestions can be combined with questionnaires for evaluation, such as collecting feedback from actual users and obtaining more real scenario requirements through questionnaires.

**Questions:**

How the 8,000 data points mentioned in the experiment were selected was not mentioned, which lacks the rigor of experimental design. It is recommended that the author describe the data mining process in detail. In addition, the specific distribution of these data for different roles is not explained. It is better to provide a chart to show it.

---

> ### Author Response · Authors · 2024-11-22
>
> We sincerely appreciate the reviewer's efforts and insightful comments to improve our manuscript. Below, we address the concerns (W) and questions (Q) raised.
>
> **[Q1] Describe the data mining process in detail.**
>
> Here is a detailed breakdown of the steps involved:
> 1. We began by extracting 301,337 real-world scenarios using the glaive-function-calling-v2 [1] and ToolLLama datasets [2] as seed conversation data, guided by the prompt detailed in Appendix A.1. To ensure the uniqueness of these scenarios, we applied a deduplication process using bge-small-en-v1.5 [3], which reduced the number of scenarios to 7,773. Subsequently, we expanded these scenarios to 15,551 using a scenario expansion prompt (Appendix A.1).
> 2. From the scenarios, we generated 15,483 compositional tasks. This involved first creating atomic tasks from the scenarios (Appendix A.2) and then generating complex compositional tasks using our heuristic strategies (Appendix A.3). To ensure the tasks' compositionality, we employed a prompt for filtering (Appendix A.3), resulting in 10,871 filtered compositional tasks. This filtering process was repeated three times, and any task that was filtered out at least once in these three iterations was deemed a bad compositional task.
> 3. Next, we generated functions corresponding to the tasks (Appendix A.4), resulting in 9,453 tasks with associated functions. Some tasks were discarded due to failures in function generation.
> 4. During the trajectory generation phase, we initially generated 8,243 trajectories using a top-down approach within a multi-agent environment (Appendix A.5). Some trajectories were discarded due to formatting issues, resulting in 8,215 data points.
> 5. We filtered out data points longer than 7 assistant turns during the trajectory generation phase. Finally, we obtained 8,000 data points, which is convenient for further analysis experiments, such as ablation studies, to align the number of data points. An example of a collected data case is provided in Appendix B.1.
>
> **[Q2] The specific distribution of these data.**
>
> We plot some key statistics of our dataset in Figure 2.
> - For the number of assistant turns, Figure 2(a) shows that it ranges from 2 to 7, and most data points involve three or more assistant response turns.
> - For the number of function calls, Figure 2(b) shows that it ranges from 1 to 6, and typically contains more than two function calls.
> - For the average number of function calls per turn, Figure 2(c) indicates an average of over one function call per turn, it indicates that parallel function calls can be occurred in every turn, and the possibility of parallel function calls decreases as the number of turns increases.
>
> For giving more insights into the data, we also add a new sunburst chart in Appendix B.3. This chart shows the distribution of the functions based on their function names. It indicates the diversity and realism of our synthesized data, and the distribution of these functions is also consistent with our daily tasks.
>
> ---
>
> [1] https://huggingface.co/datasets/glaiveai/glaive-function-calling-v2
> [2] ToolLLM: Facilitating Large Language Models to Master 16000+ Real-world APIs
> [3] https://huggingface.co/BAAI/bge-small-en-v1.5

---

### Official Review · Reviewer_UdcM · 2024-11-05

**Soundness:** 3
**Presentation:** 3
**Contribution:** 3
**Rating:** 8
**Confidence:** 5

**Summary:**

The paper discussed an approach, BUTTON, which generates synthetic compositional instruction tuning data via bottom-up instruction construction and top-down trajectory generation.

**Strengths:**

1. The research topic itself has a wide range of practical use cases.
2. The model training approach is innovative and offers a new perspective on training large language models to make multi-turn function calls.
3. The explanation of the methodology is clear and detailed. The authors trained multiple popular large language models to compare performance and validated accuracy on two benchmark datasets, providing a more comprehensive analysis.

**Weaknesses:**

In the paper, the author mentioned about “multi-turn function calling”,  “function calling” and “invoke functions” many times. Although the solution purposed could potentially solve the function calls with other additional infrastructure setups, it could be a bit misleading to the readers since the pipeline is still mocking the function call with simulated responses.

In the section 3.1 experiments setup, the author evaluated the pipeline with two different benchmark datasets. This part could be better structured.

**Questions:**

1.  It might be better to explain “function calling” a bit ahead in the abstract part to avoid confusion as the pipeline is not essentially invoking a function.
2. I would recommend to move the “related work” after the introduction to give the readers more background before introducing the methodology.
3. I would recommend to start section 3.1 experiments with a brief introduction and split them into two subsections under which you can explain the description and evaluation metrics with some bullet points.

---

> ### Author Response · Authors · 2024-11-22
>
> We sincerely appreciate the reviewer's efforts and insightful comments to improve our manuscript. Below, we address the concerns (W) and questions (Q) raised.
>
> **[W1 & Q1] Although the solution purposed could potentially solve the function calls with other additional infrastructure setups, it could be a bit misleading to the readers since the pipeline is still mocking the function call with simulated responses.**
>
> Generally, LLMs using functions can be treated as a two-stage process. The first stage is function calling, which involves generating appropriate function names and corresponding arguments. The second stage is function execution, where the function is executed and the results are obtained.
>
> For LLMs, the challenge lies in the first stage, i.e., function calling. The second stage, function execution, is mostly handled by external infrastructures. Thus, in our paper and most existing related works on LLMs with function calling, "function calling" or "invoke function" means LLMs can generate proper function names and corresponding arguments.
>
> While the function execution stage is mocked with simulated responses in the synthetic data, it's crucial to highlight that our test datasets consist of real-world functions and manually annotated queries. These real-world functions include actual implementations that can be executed with actual environments. Our experiments demonstrate that models fine-tuned with our synthetic data generalize effectively to these real-world test sets, highlighting the practical usability of our generated data.
>
> **[Q2] Move the “related work” after the introduction.**
>
> We have moved the "Related Work" section after the introduction.
>
> **[W2 & Q3] Restructure the Section 3.1 in experiments.**
>
> Due to space constraints, we have combined the related content for the two different datasets and used boldface to highlight the distinctions, aiming for maximum clarity. In future versions, if space permits, we will consider separating the descriptions of the different datasets into distinct subsections.

---

### Official Review · Reviewer_y93N · 2024-11-06

**Soundness:** 3
**Presentation:** 4
**Contribution:** 3
**Rating:** 5
**Confidence:** 3

**Summary:**

This paper introduces a new approach, BUTTON, to generate synthetic compositional instruction tuning data, which could further serve as instruction tuning data for LLMs. Compared to existing methods, this approach can engage in multi-turn function calling, which enables planning with functions in handling real-world compositional tasks.

**Strengths:**

1.	This paper first addresses the problem of planning with functions but not only use them when handling compositional tasks. The originality sounds good.
2.	This paper is well structured with good clarity.
3.	The experiment results are mostly convincing and significant compared to existing methods.

**Weaknesses:**

1.	The authors state that they differ from existing papers that ‘use functions’ by ‘plan with functions’. They are encouraged to compare with these methods. A simple way can be conducting chain-of-thought methods to decompose a compositional tasks, and then use these reference methods in each step. Can the performance still be significant enough?
2.	More experiment designs could make it more convincing. How is the performance when the task composition length, i.e. the amount of sub-tasks in one task increases? See also Question 1.

**Questions:**

1.	Does the designed heuristic strategy for compositional task construction only contain 2 or 3 sub-tasks? For example, can the sequential composition heuristic allow for an arbitrary length of sub-tasks? How is the performance according to different length of sub-tasks?
2.	It is a confusing on how function generation is conducted in Bottom-Up. The authors state that they allow for the construction of more realistic tasks, and generate functions that are likely to be called in these tasks. Is there any limitation or alignment on the function call generation? Since the function tools are specified and given in GTA and Tool-Query for evaluation, it seems that the Bottom-Up should be limited in using these given functions, rather construct new ones. Is there a conflict?
3.	The authors are encouraged to provide how prompts are designed for ablation study, i.e., how to call the single direct generation steps using one prompt.
4.	In Introduction, the bullet-in points in the last paragraph of how to solve challenges do not tackle with the challenges very well. The authors may better summarize how “compatibility” in challenge 2 and “without human supervision” in challenge 3 are tackled.
5.	Typo and grammar check, such as ‘we first extract a series of real-world scenarios from existing datasets that using external tools.' in Scenario Collection in Section 2.1.

---

> ### Author Response · Authors · 2024-11-22
>
> We sincerely appreciate the reviewer's efforts and insightful comments to improve our manuscript. Below, we address the concerns (W) and questions (Q) raised.
>
> **[W1] A simple way can be conducting chain-of-thought methods to decompose a compositional tasks, and then use these reference methods in each step. Can the performance still be significant enough?**
>
> In our existing experiments, all baselines (including GPT, Llama, and Qwen series) have employed chain-of-thought methods to decompose compositional tasks and invoke function calls based on such decomposition. The prompts used are derived from the original papers of benchmarks. Specifically, Appendix D.2 in [1] and Figure 17 in [2] detail the prompts that guide this process, breaking down a user query into multiple steps, with corresponding function calls generated at each decomposition step.
>
> **[Q1.1] Does the designed heuristic strategy for compositional task construction only contain 2 or 3 sub-tasks?**
>
> In Section 2.3 (Section 3.3 in new version), "Data Diversity", by analyzing the collected data, we note that the total number of assistant turns ranges from 2 to 7, and the number of function calls can be as high as 6. This indicates that our compositional tasks are not confined to just 2 or 3 sub-tasks.
>
> The total number of sub-tasks is not solely determined by the heuristic strategy. During function generation, multiple functions may be created for a single atomic task, allowing for the expansion of a single sub-task.
>
> For instance, Appendix B.1 provides a concrete example. The initial senario is "Machinetomachine communication implementation" and it generate an atomic task "Send a status update message from DeviceA to ServerB including the device's current timestamp".
> After using "Parallel-then-Sequential Composition", we generate parallel and sequential tasks, culminating in the final compositional task. Details are as follows:
>
> | Task | Content |
> |--------------------|-------------------------------------------------------------------------------------------------|
> | Initial Atomic Task | Send a status update message from DeviceA to ServerB including the device's current timestamp. |
> | Parallel Task      | Get the current temperature and humidity readings from DeviceA.                                 |
> | Sequential Task    | Log the status update message along with the temperature and humidity readings on ServerB's database. |
> | Composition Task    | Send a status update from DeviceA to ServerB including the current timestamp, and log this status update along with the temperature and humidity readings on ServerB's database. |
>
> While it seems that this example involves 3 sub-tasks, during the "function generation" phase, the pipeline treats the initial atomic task as two another sub-tasks: "Fetch the current timestamp from DeviceA" and "Send the status update message with the timestamp to ServerB". This results in generating functions such as ``get_current_timestamp`` and ``send_message_to_server``, thereby extending the sub-tasks and enhancing task complexity.
>
> In summary, our compositional tasks are not limited to 2 or 3 sub-tasks, as demonstrated by our analysis in Section 2.3 (Section 3.3 in new version) and the example in Appendix B.1.
>
> **[Q1.2 & W1] Can the sequential composition heuristic allow for an arbitrary length of sub-tasks? How is the performance according to different length of sub-tasks?**
>
> The sequential composition heuristic allows for sub-tasks of any length by simply repeating the process. However, during our pilot experiments, extending these strategies too far could negatively affect final performance. We believe this occurs because the current BUTTON framework already generates complex and diverse compositional tasks requiring multi-turn function calls (as noted in response to [Q1.1]). Repeated application of these strategies often results in overly complex tasks with limited practical value.
>
> Despite not extending the simple heuristic, our collected data remains diverse (as mentioned in Section 2.3 (Section 3.3 in new version)). As demonstrated in our experiments, the LLMs fine-tuned with our generated data effectively handle tasks with different length of sub-tasks. For instance, the labeled number of turns of queries in GTA are ranged from 2 to 8 (Figure 3(a) in [1]) and the average number of turns is 5 in Tool-Query (Table 3 in [2]). In our evaluation on Tool-Query, tasks needing fewer or more than 4 turns were categorized as easy and hard, respectively (Table 3). Our model excelled in both categories, indicating its capability to handle tasks with different length of sub-tasks.

---

> ### Author Response · Authors · 2024-11-22
>
> **[Q2] Is there any limitation or alignment on the function call generation? Since the function tools are specified and given in GTA and Tool-Query for evaluation, it seems that the Bottom-Up should be limited in using these given functions, rather construct new ones. Is there a conflict?**
>
> We do not impose restrictions or alignments between the tools in our generated dataset and those in the evaluation datasets because we don't assume the same tool set during training and testing. Our framework is designed to enhance generalization in function-calling abilities, enabling models to effectively handle unseen functions. This aligns with the primary goal of instruction tuning for LLMs, which is to improve their generalization to new tasks.
>
> By incorporating question-specific available functions (i.e., tools) into the system prompt for each user question during training or testing (as demonstrated by the case in Appendix B.1), it helps *LLMs learn to understand and leverage given available functions, rather than memorizing all functions during training*. This setting is consistent with the standard practice in function calling for LLMs [1,2].
>
> In summary, there is no limitation or alignment on the function generation, as LLMs fine-tuned by our generated data can generalize to unseen functions during evaluation.
>
> **[Q3] The authors are encouraged to provide how prompts are designed for ablation study.**
>
> Certainly, we have included the prompts used for the ablation study in Appendix C.3.
>
> **[Q4] The authors may better summarize how "compatibility" in challenge 2 and "without human supervision" in challenge 3 are tackled.**
>
> In addressing Challenge 2, "How to ensure the compatibility of an instruction with its functions", our approach generates compositional tasks bottom-up from atomic tasks. This ensures that sub-tasks are integrated into the function generation process, enhancing compatibility with the overall compositional task. Without this approach, function generation would only consider the compositional task, missing insights from the atomic sub-tasks. Our ablation study demonstrates the effectiveness of this method: the "w/o bottom-up" results show that without incorporating atomic sub-tasks leads to less compatible function generation and lower performance.
>
> In addressing Challenge 3, "How to simulate high-quality multi-turn function calling trajectories without human supervision", in the top-down phase, our approach employs multi-agent systems to simulate trajectories in a completely end-to-end manner, eliminating the need for human supervision. Our ablation study highlights the importance of this method: the "w/o top-down" setting shows that bypassing the multi-agent environment and generating trajectories directly results in significantly lower performance. This clearly demonstrates the effectiveness of our top-down phase without human supervision.
>
> **[Q5] Typo and grammar check.**
>
> We have revised the manuscript to address the typos and grammar errors.
>
> ---
>
> > [1] GTA: A Benchmark for General Tool Agents
> [2] AgentBoard: An Analytical Evaluation Board of Multi-turn LLM Agents

---

> ### Author Response · Authors · 2024-12-03
>
> Dear Reviewer y93N:
>
> We sincerely appreciate the constructive comments and insightful suggestions you have provided for our work. With the discussion period ending soon on December 2nd, AoE, we kindly encourage you to engage in the ongoing discussion and share any further insights or clarifications in response to our work. Thank you for your time and consideration. We look forward to your response.

---

### Meta-Review · Area_Chair_FUDn · 2024-12-20

**Metareview:**

(a) Summary: This paper introduces a new approach, BUTTON, to generate synthetic compositional instruction tuning data, which could further serve as instruction tuning data for LLMs.
(b) Strengths: The paper is generally well-written and easy to follow. The proposed approach seems interesting and reasonable. The experimental results seem to support the authors' claims.
(c) Weaknesses: The reviewers pointed out a few concerns and issues. Most of them are clarification questions.
(d) Although some of the reviewers did not reply to the authors' rebuttal, I think most of the concerns and comments have been addressed.

**Additional Comments On Reviewer Discussion:**

The authors provided additional explanations and details to address the reviewers' concerns in their rebuttal. Only one reviewer replied to the rebuttal. I think that most of the concerns and comments have been addressed.

---

### Decision · Program_Chairs · 2025-01-22

Accept (Poster)